# CNK2 promotes cancer cell motility by mediating ARF6 activation downstream of AXL signalling

Guillaume Serwe[1,2,5], David Kachaner[1,5], Jessica Gagnon[1,2,5], Cédric Plutoni[1], Driss Lajoie[1], Eloïse Duramé[1,2], Malha Sahmi[1], Damien Garrido [1], Martin Lefrançois[1], Geneviève Arseneault[1], Marc K. Saba-El-Leil[1], Sylvain Meloche [1,2,3], Gregory Emery [1,2,4] & Marc Therrien [1,2,4] ✉

Cell motility is a critical feature of invasive tumour cells that is governed by complex signal transduction events. Particularly, the underlying mechanisms that bridge extracellular stimuli to the molecular machinery driving motility remain partially understood. Here, we show that the scaffold protein CNK2 promotes cancer cell migration by coupling the pro-metastatic receptor tyrosine kinase AXL to downstream activation of ARF6 GTPase. Mechanistically, AXL signalling induces PI3K-dependent recruitment of CNK2 to the plasma membrane. In turn, CNK2 stimulates ARF6 by associating with cytohesin ARF GEFs and with a novel adaptor protein called SAMD12. ARF6-GTP then controls motile forces by coordinating the respective activation and inhibition of RAC1 and RHOA GTPases. Significantly, genetic ablation of CNK2 or SAMD12 reduces metastasis in a mouse xenograft model. Together, this work identifies CNK2 and its partner SAMD12 as key components of a novel pro-motility pathway in cancer cells, which could be targeted in metastasis.

Scaffold proteins confer selectivity and spatiotemporal regulation to signal transduction by assembling molecular complexes and targeting them to specific subcellular locations[1]. The fine-tuning they provide is necessary to ensure normal cell responses to signalling cues and loss-of-function or altered expression of scaffolds can therefore contribute to disease including cancer[2,3]. Scaffolds of the Connector enhancer of kinase suppressor of RAS (CNK) family are evolutionary conserved proteins whose functions in signal transduction are poorly characterized. CNK was discovered through genetic studies in *Drosophila* as a positive regulator of the receptor tyrosine kinase (RTK)-induced RAS/MAPK pathway during eye development[4,5]. The human genome contains four paralogous *CNKSR* genes (*CNKSR1*, *CNKSR2*, *CNKSR3*, *IPCEF1*) which encode the proteins CNK1, CNK2, CNK3, and IPCEF1 (ref. 6). CNKs are composed of protein- and phospholipid-interaction domains

including a sterile alpha motif (SAM), a PSD-95/DLG1/ZO-1 (PDZ), and a pleckstrin-homology (PH) domain (Fig. 1a). They also contain two conserved regions, which mediate protein-protein interactions: an N-terminal Conserved Region In CNKs (CRIC) and, except for CNK3, a C-terminal Conserved Region Among Chordates (CRAC). The physiological functions of human CNKs and their roles in disease are largely unknown. However, loss-of-function mutations in the *CNKSR2* locus, which is primarily expressed in neuronal tissues[7], are associated with the occurrence of X-linked intellectual disability and epilepsy[8–10]. Consistently, *CNKSR2* knockout (KO) in mice causes neural hyperactivity and behavioural phenotypes that are characteristic of epilepsy-aphasia syndromes[11]. In contrast, KO models of *CNKSR1*, *CNKSR3*, or *IPCEF1* are not yet available, limiting our understanding of their biological functions.

[1]Institute for Research in Immunology and Cancer, Université de Montréal, Montréal, QC, Canada. [2]Molecular Biology Program, Faculty of Medicine, Université de Montréal, Montréal, QC, Canada. [3]Department of Pharmacology and Physiology, Faculty of Medicine, Université de Montréal, Montréal, QC, Canada. [4]Department of Pathology and Cell Biology, Faculty of Medicine, Université de Montréal, Montréal, QC, Canada. [5]These authors contributed equally: Guillaume Serwe, David Kachaner, Jessica Gagnon. ✉e-mail: marc.therrien@umontreal.ca

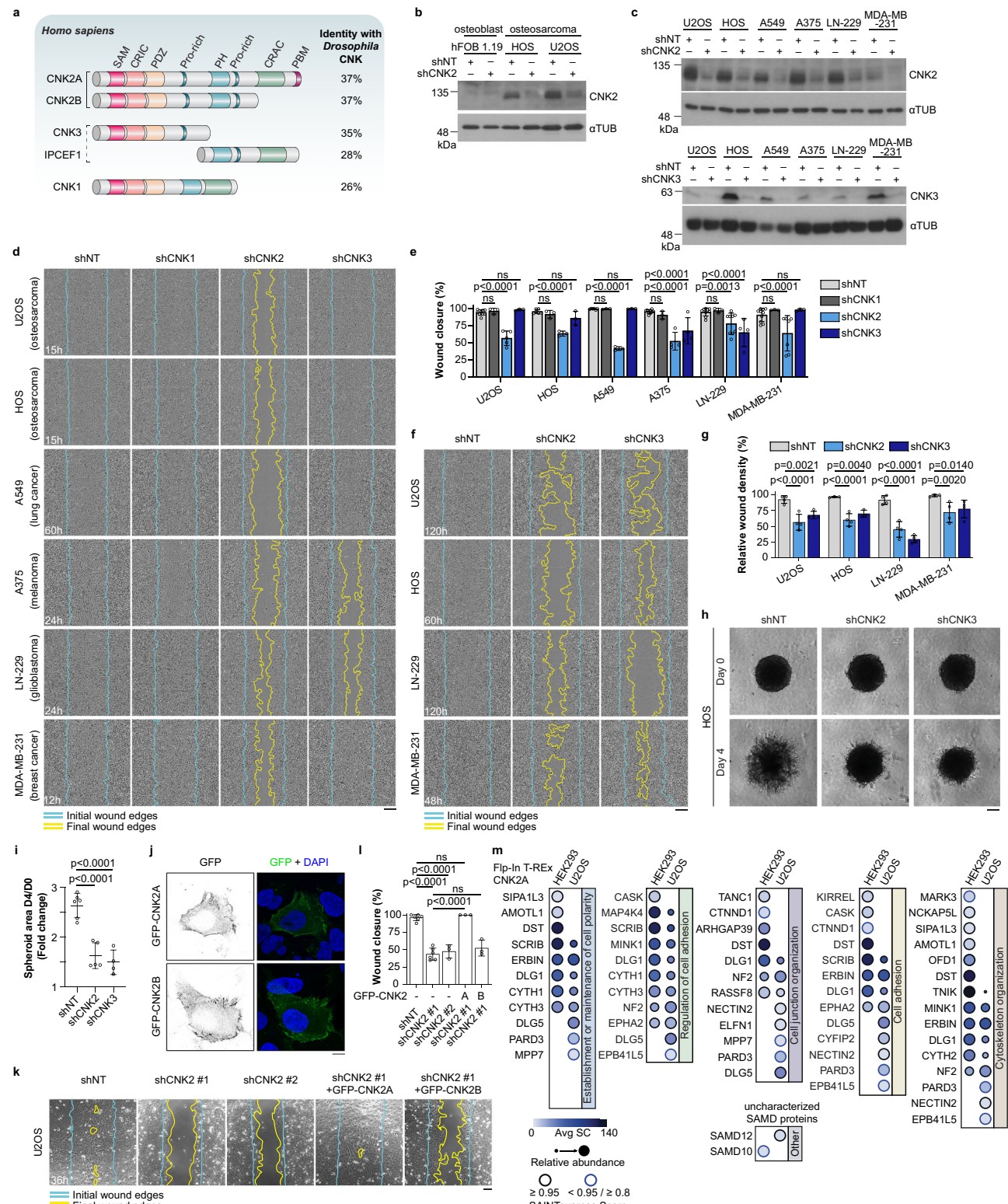

Although human CNK1 and CNK2 can interact with RAF kinases[12,13] like *Drosophila* CNK, evidence demonstrating their requirement for RAS/MAPK signalling remains sparse[14]. However, mammalian CNKs have been linked to signalling pathways controlled by other classes of GTPases, including members of the ARF and RHO subfamilies[15–17]. ARF and RHO GTPases are key regulators of vesicular trafficking, cytoskeletal remodelling, and cell motility, and are implicated in oncogenesis[18–21]. Yet, despite their connections to these GTPases, the cellular functions of CNKs remain largely uncharacterized.

In this work we used a proteomics approach to obtain a global view of CNK functions in human cells and found that the proximal interactomes of CNK2 and CNK3 display a strong signature for biological processes involved in cell morphogenesis. Considering that morphogenetic remodelling is a defining aspect of tumour cell dissemination and cell motility, the interactome signature of CNKs led us to discover that CNK2 is required for cell migration, invasion, and metastasis. Intriguingly, despite being predominantly restricted to healthy neuronal tissues, we found that CNK2 is expressed and confers

**Fig. 1 | CNK2 knockdown delays cell migration and invasion of multiple cancer cell lines. a** Schematic representation of human CNK proteins. CNK homologs are depicted with their conserved domains or regions (see text for description). Bracketed CNK2 proteins indicate two splicing variants encoded by the *CNKSR2* gene. CNK2A contains a PDZ domain-binding motif (PBM) at its C-terminus. The dashed bracket denotes that CNK3 and IPCEF1 might be encoded by the same gene. CNK2 is the closest ortholog to *Drosophila* CNK based on sequence identity. **b** CNK2 expression is detected in two osteosarcoma cell lines (U2OS, HOS) but not in osteoblast cells (hFOB 1.19). shRNA-mediated depletion of CNK2 confirmed the specificity of our custom-made antibody. Levels of α-tubulin (αTUB) were used as loading controls. **c** Immunoblots confirming shRNA-mediated depletion of endogenous CNK2 and CNK3 in six cancer cell lines. **d** shRNA-mediated depletion of CNK2 delayed migration of six cancer cell lines and knockdown of CNK3 reduced migration of A375 and LN-229 in wound healing assays. Blue lines delineate the initial wound edges and yellow lines delineate the migration or invasion front at the indicated time point. The same colour-coded lines were used for all subsequent wound healing assays. **e** Quantification of the migration assays shown in **d** at the indicated time points for each cell line. **f** Depletion of CNK2 or CNK3 delayed invasion of the indicated cell lines used in Matrigel-based wound healing assays. **g** Quantification of the invasion assays shown in (**f**) at the indicated time points for each cell line. **h** Depletion of CNK2 or CNK3 by shRNA reduced 3D invasion of HOS cell spheroids into Matrigel. **i** Quantification of the invasion assays shown in (**h**). **j** GFP-CNK2A and GFP-CNK2B are mainly detected at the plasma membrane in U2OS cells. **k** Expression of shRNA-insensitive GFP-CNK2A but not GFP-CNK2B rescued the migration delay observed in CNK2-depleted U2OS cells in wound healing-like assays. **l** Quantification of the migration assays shown in (**k**). **m** Comparative dot plot of the proximal interactors identified by BioID for CNK2A in HEK293 (data also presented in Supplementary Fig. 1d) and U2OS Flp-In T-REx cells. The proximal interactors are grouped into Biological Processes (Gene Ontology) of interest. All statistical analyses were performed on data from at least three independent experiments. Two-way ANOVA was used in (**e**) and (**g**), and one-way ANOVA was used in (**i**) and (**l**). Error bars correspond to mean values ± SD; ns, not significant ($P > 0.05$). Scale bar: **d**, **f** 200 μm, **h**, **k** 100 μm, **j** 10 μm. Source data are provided as a Source Data file.

enhanced cell motility in several non-neuronal cancer cells lines. Using osteosarcoma cells to decipher the underlying mechanism, we discovered a CNK2-dependent signalling axis, triggered by the RTK AXL, that coordinates the activity of ARF6, RAC1, and RHOA GTPases. Overall, our work uncovers an unanticipated function of CNK2 in regulating cancer cell motility and identifies a novel signalling pathway that could be targeted to restrain metastasis.

## Results

### Defining the proximal interactome of CNKs in human cells

The functions of human CNK proteins remain poorly defined. To determine through an unbiased approach which cellular processes CNK1, 2, and 3 could be involved in, we used the biotin identification (BioID) method to uncover their proximal interactomes in Flp-In T-REx HEK293 cells expressing FLAG-BirA*-tagged CNK proteins (Supplementary Fig. 1a–e). For CNK2 BioID, we used the CNK2A isoform as "bait", which contains a C-terminal region harbouring the CRAC and a PDZ-binding motif (PBM) (Fig. 1a). The BioID screen identified 75 confident proximal interactors for these CNK proteins (Supplementary Fig. 1d; Supplementary Data 1). Among them were known CNK interactors including CYTH1, 2, and 3, which were specific to CNK1 and CNK2A, and TNIK, ARHGAP39 (Vilse), and SCRIB which are known interactors of CNK2A[15,22]. Interestingly, the proximal interactomes of CNK2A and CNK3 contained 18 proteins in common which were not shared with CNK1, suggesting a functional divergence between CNK proteins. This is consistent with the closer sequence identity between CNK2 and CNK3 compared to CNK1 (Supplementary Fig. 1f). Clustering proximal partners based on their gene ontology in biological processes revealed a significant enrichment in the CNK2A and CNK3 proximal interactomes for processes that contribute to cell morphogenesis[23], including cell junction organization, cytoskeleton organization, establishment or maintenance of cell polarity, cell adhesion, and regulation of cell adhesion. As these processes are deregulated during epithelial-mesenchymal transition (EMT) and contribute to the acquisition of motile and invasive phenotypes by cancer cells[24], we hypothesized that CNK2 and CNK3 could be involved in regulating cell motility.

### CNK2 promotes cancer cell migration and invasion

To identify cancer cell models that could be used to investigate the role of CNKs in motility, we used RT-qPCR to measure the expression of *CNKSR1, 2*, and *3* in a panel of 39 human cancer cell lines of diverse origins (Supplementary Data 2). We detected CNK1 and CNK3 encoding transcripts in all cell lines, and while CNK1 levels were low in most of them, CNK3 was more abundant. As expected, CNK2 transcript levels were high in neuronal cancer cells, but surprisingly, it was also expressed in several non-neuronal cell lines. Notably, we detected the

CNK2 protein in two osteosarcoma cell lines despite its lack of expression in normal osteoblast cells (Fig. 1b).

To test the role of CNK2 and CNK3 in cancer cell motility, we conducted wound healing migration assays using six cancer cell lines that co-expressed both genes (Supplementary Data 2) and that were capable of collective cell migration, namely, U2OS (osteosarcoma), HOS (osteosarcoma), A549 (lung adenocarcinoma), A375 (melanoma), LN-229 (glioblastoma), and MDA-MB-231 (breast adenocarcinoma) cells. Compared to cells expressing a non-target shRNA (shNT), CNK2 depletion (shCNK2, also referred to as shCNK2 #1 in subsequent figures) delayed cell migration of all six cell lines, while CNK3 depletion delayed migration of A375 and LN-229 cells (Fig. 1c–e). CNK1 expression was barely detectable in these cell lines and its knockdown did not affect cell migration (Fig. 1d, e; Supplementary Fig. 2a; Supplementary Data 2). To confirm that delayed wound closure was not due to reduced proliferation or cell death, we conducted viability assays which revealed that depletion of CNK1, 2, or 3 did not significantly affect cell proliferation or viability (Supplementary Fig. 2b). Since cancer cell dissemination involves invasion into the extracellular matrix, we tested the requirement for CNK2 and CNK3 during this process using Matrigel-based 2D wound healing and 3D spheroid invasion assays. Four of the six cell lines that we used for migration assays were amenable to 2D invasion (U2OS, HOS, LN-229, MDA-MB-231), and knockdown of CNK2 or CNK3 in each of them delayed invasion (Fig. 1f, g). Additionally, we used HOS cells as a model for spheroid assays and found that depletion of CNK2 or CNK3 also impaired 3D invasion (Fig. 1h, i). Overall, our migration and invasion assays using diverse cancer cell lines identify CNK2 and CNK3 as regulators of cell motility.

Given that depletion of CNK2 had the most widespread effect across the tested cell lines, we aimed to characterize the mechanisms by which it functions during cell migration. Using U2OS cells as a model, we conducted rescue experiments to identify the functional CNK2 isoform (Fig. 1a). While exogenous CNK2A and CNK2B both localized in the cytoplasm and were enriched at the plasma membrane[17] (Fig. 1j), the migration delay caused by knockdown of CNK2 was rescued by expression of shRNA-insensitive GFP-CNK2A, but not GFP-CNK2B (Fig. 1k, l; Supplementary Fig. 2c; Supplementary Movie 1), indicating that the CNK2A C-terminal region is required for its function. This result also shows that delayed migration is not due to an off-target effect of the shRNA, and supporting this notion, CNK2 knockdown with a second shRNA also delayed cell migration (Fig. 1k, l; Supplementary Fig. 2c). We then set out to define the proximal interactome of CNK2A in a more relevant cellular context for investigating cell migration by conducting a BioID in Flp-In T-REx U2OS cells (Fig. 1m; Supplementary Fig. 3a–d; Supplementary Data 3). 56% of the recovered proximal interactors were also observed in HEK293 cells,

and accordingly, the CNK2A proximal interactome in U2OS cells also displayed a clear bias for interactors involved in cell morphogenetic processes and was thus consistent with a role in cell motility. Importantly, this approach identified a network of candidates (Supplementary Fig. 3e) that could be involved in mediating the function of CNK2A in cell motility.

## CNK2 regulates RAC1 and RHOA GTPases during migration

RHO GTPases regulate crucial steps of the migration cycle, including the formation of polarized protrusions, adhesion to the substrate, and the generation of contractile forces[20,25]. RAC1, RHOA, and CDC42 are the most characterized RHO GTPases in the regulation of motility[26,27], and we therefore sought to determine whether the migration delay caused by CNK2 depletion could be a consequence of deregulation of these GTPases. We conducted pull-down experiments to measure the GTP-loaded state of RAC1, RHOA, and CDC42 during cell migration[28] and found that CNK2-depleted cells contained significantly lower levels of RAC1-GTP and higher levels of RHOA-GTP compared to control cells (Fig. 2a, b). The amount of GTP-bound CDC42, however, was unchanged by CNK2 knockdown (Fig. 2c). Additionally, CNK1 knockdown had no impact on the GTP-bound levels of these GTPases, while CNK3-depleted cells displayed a statistically significant but modest decrease of CDC42-GTP levels (Fig. 2a–c). Considering the role of *Drosophila* CNK in RAS/MAPK signalling, we also tested whether human CNKs modulate RAS-GTP levels during migration, but this was not the case (Supplementary Fig. 4a). Taken together, these results suggest that impaired cell migration caused by CNK2 knockdown is a result of decreased RAC1 activity and increased RHOA activity.

RAC1 is a crucial regulator of actin polymerization at the front of migrating cells and drives the formation of lamellipodia[25–27]. To assess whether CNK2 regulates the formation of lamellipodia, we scored the shape of protrusions and measured protrusion width in control cells and CNK2-depleted cells. We found that significantly fewer cells display fan-like lamellipodia at the migration front in the wound healing assay following CNK2 knockdown (Supplementary Fig. 4b, c). Instead, cells lacking CNK2 often display protrusions that are thinner, and they also possess a greater number of protrusions, indicating a defect in forming a lamellipodium (Supplementary Fig. 4b, d, e). These defects were rescued by exogenous expression of GFP-CNK2A but not GFP-CNK2B, indicating that the CNK2A isoform promotes lamellipodia formation during cell migration (Supplementary Fig. 4b–e). In agreement with these findings, migrating CNK2-depleted cells that were stained with phalloidin to mark F-actin also displayed fewer fan-like protrusions (Supplementary Fig. 4f, g).

During cell migration, activated RHOA stimulates its effector kinase ROCK, which phosphorylates myosin light chain 2 (MLC2) and drives the formation of contractile actin stress fibres and mature focal adhesion complexes (FAs)[29–31]. To determine whether CNK2 depletion increases signalling downstream of RHOA, we assessed the abundance and localization of stress fibres and FAs in migrating cells. Using phosphorylated MLC2 (pMLC2) and phalloidin staining as markers for stress fibres, we found that CNK2-depleted cells displayed an increase in the amount of ventral contractile stress fibres and their redistribution from the cell periphery to the centre of the cell (Fig. 2d; Supplementary Fig. 4h). Additionally, staining of the mature FA component zyxin revealed that cells lacking CNK2 possessed larger and more abundant mature FAs, which were also localized more centrally than in control cells (Fig. 2e). The impact of CNK2 depletion on stress fibres and mature FAs was largely restored by re-expression of GFP-CNK2A, but not GFP-CNK2B, further indicating that RHOA signalling is negatively regulated by CNK2A and demonstrating the functional relevance of the C-terminal extension of CNK2A (Fig. 2d, e; Supplementary Fig. 4h). Furthermore, inhibition of ROCK using Y-27632 also rescued the cell migration delay caused by CNK2 knockdown (Fig. 2f, g) and led to a decrease of F-actin staining and pMLC2 levels (Fig. 2h;

Supplementary Fig. 4i), indicating that the accumulation of stress fibres in cells lacking CNK2 depends on the kinase activity of ROCK. Interestingly, both CNK2-depleted cells and control cells treated with Y-27632 displayed prominent membrane ruffles, which are dependent on RAC1 activity[32] (Fig. 2h). Since RAC1 and RHOA activity is mostly mutually antagonistic[33] (see below), this observation is consistent with ROCK inhibition. Overall, these data establish CNK2 as a positive regulator of RAC1 and as a negative regulator of RHOA during cell migration.

## Cytohesin-1 and −3, and SAMD12 are functional CNK2 binding partners

To determine how CNK2A regulates cell migration, we hypothesized that, as a scaffold, it forms protein interactions that are required for its function. Of the 33 proximal interactors specific to CNK2A in HEK293 cells (Supplementary Fig. 1d), 14 were also detected in U2OS cells including Misshapen family kinases (MAP4K4, TNIK, MINK1), cytohesin family ARF GEFs (CYTH1, 2, 3), and the scaffold proteins NF2 and SCRIB, all of which were high confidence hits (Fig. 1m). We confirmed that most of these proximal interactors co-immunoprecipitate (co-IP) with CNK2A in Flp-In T-REx U2OS cells (Supplementary Fig. 5a), and endogenous co-IP demonstrated that MAP4K4, SCRIB, NF2, and cytohesins interact specifically with CNK2 in U2OS cells (Fig. 3a). Interestingly, we also identified SAMD12 (Fig. 1m), a human ortholog of the *Drosophila* CNK binding partner Hyphen (HYP), which is essential for RAS/MAPK signalling[34,35]. HYP is composed of a single SAM domain and functions as an adaptor protein but the molecular role of SAMD12 is uncharacterized. Through co-IP we confirmed that FLAG-tagged SAMD12 interacts with exogenous CNK2A in U2OS cells (Fig. 3b). Of the interactors that co-IP with endogenous or exogenous CNK2, depletion of MAP4K4, TNIK, CYTH1, CYTH3, or SAMD12 by shRNA delayed U2OS cell migration (Fig. 3c; Supplementary Fig. 5b–d), identifying them as candidate functional interactors. Since the biological functions of SAMD12 were unknown, we conducted additional motility assays to characterize this protein. We discovered that, like CNK2 knockdown, depletion of SAMD12 impaired cell migration and 2D invasion in multiple cancer cell lines, and 3D invasion in HOS cells (Fig. 3d-i; Supplementary Fig. 5d), establishing SAMD12 as a positive regulator of cell motility.

To determine the functional relevance of CNK2A interactions, we introduced mutations into a shRNA-insensitive GFP-CNK2A construct that disrupted binding with its partners and assessed their impact in cell migration rescue experiments. We tested six different mutations (Fig. 3j): (1) a double-charge reversal of two conserved glutamic acid residues in the SAM domain (E64R/E68R) that abolished its interaction with FLAG-SAMD12 (Fig. 3b) and whose corresponding mutation impairs *Drosophila* CNK/HYP binding[36]; (2) a three-residue deletion in the CRIC (ΔQ172/Q173/D174) that perturbed binding with MAP4K4 (Fig. 3k) and whose corresponding deletion causes a loss-of-function in *Drosophila* CNK[5,37]; (3) a single mutation in the CRAC (L955P), the known binding region for cytohesins[15,17], that prevented its interaction with CYTH1 and CYTH3 (Fig. 3l); (4) a triple mutation of three conserved residues in the CRAC (T961A/L962A/K963A) that specifically impaired binding with CYTH2 (Fig. 3l), (5) deletion of the PBM (ΔETHV) which prevents binding with SCRIB[17] (Supplementary Fig. 5e). Lastly, to assess the function of the CNK2A PDZ domain, we introduced (6) a double mutation in its hydrophobic pocket (G228S/M229S) that is predicted to disrupt binding with PBMs[38]. Strikingly, CNK2A constructs carrying the mutation(s) that impacted SAMD12 (E64R/E68R) or CYTH1 and CYTH3 (L955P) binding were unable to rescue cell migration and velocity in CNK2-depleted cells (Fig. 3m, n; Supplementary Fig. 5f, g; Supplementary Movie 2). Loss-of-function was not due to mislocalization of the mutants since they localized in the cytoplasm and to sites of cell-cell contact like CNK2A WT (Supplementary Fig. 5h). These results suggest that the function of CNK2 in cell migration

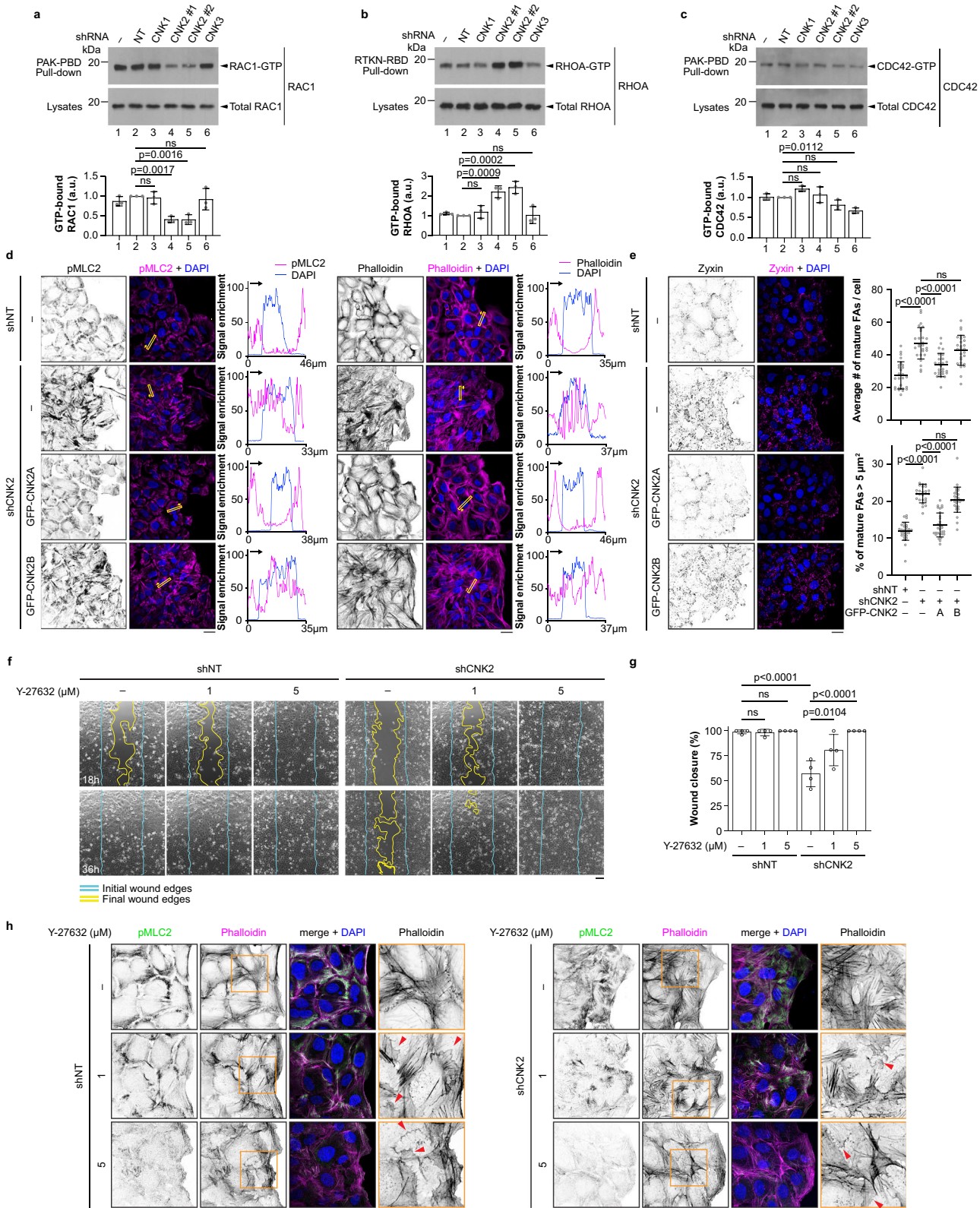

requires its interaction with SAMD12 and the ARF GEFs CYTH1 and CYTH3.

## CNK2 stimulates ARF6 GTPase upstream of RAC1 and RHOA

Since cytohesins are GEFs for ARF GTPases, we asked whether CNK2 and its functional interactors regulate ARFs during cell migration. ARF1 and ARF6 have been shown to promote cell migration[39,40], and we

therefore used pull-downs to test the impact of CNK2 knockdown on their activation state. We found that depletion of CNK2 with two different shRNAs caused a decrease of ARF6-GTP levels but had no impact on ARF1 activity (Fig. 4a; Supplementary Fig. 6a). Depletion of the ARF GEFs CYTH1 or CYTH3 also caused a reduction of ARF6-GTP, but surprisingly CYTH2 depletion had no impact (Fig. 4b). Strikingly, knockdown of SAMD12 also caused a strong reduction of ARF6-GTP

**Fig. 2 | CNK2 is a positive regulator of RAC1 and a negative regulator of RHOA during migration. a–c** Depletion of CNK2 in migrating U2OS cells decreased RAC1-GTP levels and increased RHOA-GTP levels. GTP-bound RAC1 (**a**), RHOA (**b**), and CDC42 (**c**) were precipitated with PAK-PBD or Rhotekin-RBD beads as indicated. Bar graphs indicate the normalized ratio of GTP-bound GTPase to the amount of total proteins. **d** Migrating U2OS cells were stained for pMLC2 (left panel) or with phalloidin (right panel). Cells expressed a non-target shRNA or a shRNA targeting CNK2. In CNK2-depleted cells, shRNA-insensitive GFP-CNK2A or GFP-CNK2B were overexpressed as indicated. Line scans indicate the immunofluorescence signal enrichment for pMLC2, phalloidin, and DAPI in the yellow-boxed regions. Yellow arrows indicate the orientation of the line scans on the x-axis. **e** Images of U2OS cells stained for Zyxin. Cells were depleted of CNK2 and shRNA-insensitive constructs were overexpressed as indicated. Quantification of the average number of

mature focal adhesions per cell (>1 μm²) and the percentage of mature focal adhesions >5 μm² is shown. At least 25 images from three independent experiments were used for quantification. **f** Inhibition of ROCK with Y-27632 (1 or 5 μM) rescued the migration delay observed in CNK2-depleted U2OS cells in wound healing-like assays. Images were taken at the indicated time points. **g** Quantification at 36 h of the migration assays shown in (**f**). **h** Migrating U2OS cells were depleted of CNK2 and treated with the ROCK inhibitor Y-27632 (1 or 5 μM) or DMSO and stained for pMLC2 and with phalloidin. The area within the orange square is magnified on the right. Red arrows point to membrane ruffles. All statistical analyses were one-way ANOVA and were performed on data from three independent experiments. Error bars correspond to mean values ± SD; ns, not significant ($P > 0.05$). Scale bar: **d**, **e**, **h** 20 μm, **f** 100 μm. Source data are provided as a Source Data file.

levels (Fig. 4b). Importantly, in contrast to CNK2A WT, expression of GFP-CNK2A constructs carrying the mutations that impair CYTH1 and CYTH3 (L955P) or SAMD12 (E64R/E68R) binding were unable to restore ARF6-GTP levels in CNK2-depleted cells (Fig. 4c), indicating that binding between CNK2A and these interactors is necessary to promote ARF6 activity. We confirmed that ARF6 is required for cell migration in U2OS cells as its depletion by shRNA delayed migration in the wound healing assay (Supplementary Fig. 6b, c). Additionally, the migration delay caused by CNK2 depletion was rescued by over-expression of FLAG-CYTH1 WT, but not the nucleotide exchange function-defective mutant (E157K)[41] (Fig. 4d, e; Supplementary Fig. 6d), consistent with the notion that reduced ARF6 activity contributes to the impaired migration of CNK2-depleted cells. In line with this finding, cell migration of CNK2-knockdown cells was also rescued by exogenous expression of constitutively active (Q67L)[42,43] and fast-cycling (T157A)[44] ARF6 mutants, but not by the dominant negative (T27N)[42,43] mutant (Fig. 4f, g; Supplementary Fig. 6e). Overall, these results indicate that CNK2 promotes cell migration by increasing the levels of active ARF6.

ARF6 regulates cell migration in MDCK cells by promoting the activation of RAC1 at the leading edge of the cell[45,46]. Additionally, overexpression of a constitutively active ARF6 mutant (Q67L) causes a loss of stress fibres in CHO cells, suggesting that ARF6 antagonizes RHOA[47]. We hypothesized that CNK2-dependent regulation of RAC1 and RHOA is mediated by ARF6, and we therefore asked whether ARF6 depletion would have the same impact on RAC1 and RHOA activity as CNK2 knockdown. We also asked whether CNK2 binding partners that were necessary for cell migration, namely CYTH1 and CYTH3, and SAMD12, could also regulate RAC1 and RHOA. We conducted pull-down assays to measure RAC1- and RHOA-GTP levels in U2OS cells during migration and found that shRNA-mediated knockdown of ARF6, CYTH1, CYTH3, or SAMD12 caused a reduction of RAC1-GTP and an increase of RHOA-GTP (Fig. 4h–j), similar to CNK2 knockdown (Fig. 2a, b). As expected, depletion of CYTH2, which did not impact cell migration or ARF6 activity (Figs. 3c; 4b), had no significant effect on RAC1-GTP or RHOA-GTP levels (Fig. 4i, j). These results indicate that, like CNK2, its binding partners SAMD12, CYTH1 and CYTH3, as well as the downstream ARF6, regulate the activity of RAC1 and RHOA during migration.

RAC1 and RHOA control different aspects of cytoskeletal dynamics during migration and their functions are mutually antagonistic[33]. We used pull-downs to test whether such antagonism exists between RAC1 and RHOA activity during cell migration in U2OS cells. Depletion of RAC1 led to an increase of RHOA-GTP levels and depletion of RHOA had the same effect on RAC1-GTP (Fig. 4k, l), indicating that these GTPases are indeed antagonistic to each other. Additionally, since we hypothesized that CNK2-dependent ARF6 activation occurs upstream of RAC1 and RHOA, we expected that ARF6 activity would not be modulated by RAC1 and RHOA. In agreement with this, RAC1 depletion had no impact on ARF6 activity (Fig. 4m) even though it delayed cell migration (Supplementary Fig. 6b, c),

indicating that RAC1 functions downstream of ARF6. RHOA depletion caused a slight increase of ARF6-GTP levels (Fig. 4m) and did not delay cell migration (Supplementary Fig. 6b, c). Overall, these results indicate that ARF6 functions upstream of RAC1 and RHOA during cell migration, and strongly suggest that CNK2-dependent regulation of RAC1 and RHOA is mediated by ARF6.

## CNK2 operates at the plasma membrane

Since CNK proteins except CNK3 possess a phosphoinositide-binding PH domain, which can mediate plasma membrane recruitment, we hypothesized that the function of CNK2 in cell migration would depend on its membrane localization. Through cell fractionation experiments, we determined that endogenous CNK2 is enriched at the plasma membrane and is also detected in the cytoplasm (Fig. 5a). This finding was corroborated by immunofluorescence, which showed a colocalization between CNK2 and the adherens junction marker N-cadherin (Fig. 5b). In some instances, CNK2 was also detected at the leading edge of migrating cells (Fig. 5b). To determine whether the presence of CNK2A at the plasma membrane is necessary for its function, we substituted tryptophan 591 to alanine (W591A) in the CNK2A PH domain, which is a conserved residue among PH domain-containing proteins that is necessary for protein-phosphoinositide interactions[48]. Accordingly, this mutation abolished the detection of GFP-CNK2A at the plasma membrane (Fig. 5c; Supplementary Fig. 7a). In contrast to GFP-CNK2A WT, expression of GFP-CNK2A W591A in cells depleted of endogenous CNK2 failed to rescue cell migration (Fig. 5d–f; Supplementary Movie 3), demonstrating that the membrane localization of CNK2 is critical for its function in cell migration.

We also observed that overexpression of GFP-CNK2A WT increased wound healing migration and cell velocity (Supplementary Fig. 7b–e; Supplementary Movie 4) and led to elevated RAC1-GTP and ARF6-GTP levels, and lower RHOA-GTP levels (Fig. 5g–i). Cells over-expressing GFP-CNK2A WT also displayed reduced filamentous actin staining and a noticeable alteration of cell shape (Supplementary Fig. 7a, f). In contrast, overexpression of GFP-CNK2A W591A or GFP-CNK2B had no effect on cell migration, GTPase activity, or F-actin distribution and cell morphology (Fig. 5g–i; Supplementary Fig. 7a–f; Supplementary Movie 4), indicating that the ability of CNK2A to regulate ARF6, RAC1, and RHOA during migration requires its membrane localization.

We next asked whether CNK2 forms a complex with cytohesins and ARF6 at the plasma membrane. Through co-IP experiments, we found that ARF6-V5 interacts with GFP-CNK2A WT and that their association is strongly disrupted by the CNK2A W591A mutation (Fig. 5j), indicating that they interact predominantly at the plasma membrane. Additionally, binding between ARF6-V5 and GFP-CNK2A WT was increased in the presence of FLAG-CYTH1 (Fig. 5j), suggesting that CYTH1 mediates their interaction. By immunofluorescence, we observed that FLAG-CYTH1 was predominantly cytoplasmic and that co-expression with GFP-CNK2A WT, but not with GFP-CNK2A W591A or GFP-CNK2B, caused a drastic increase in the percentage of cells

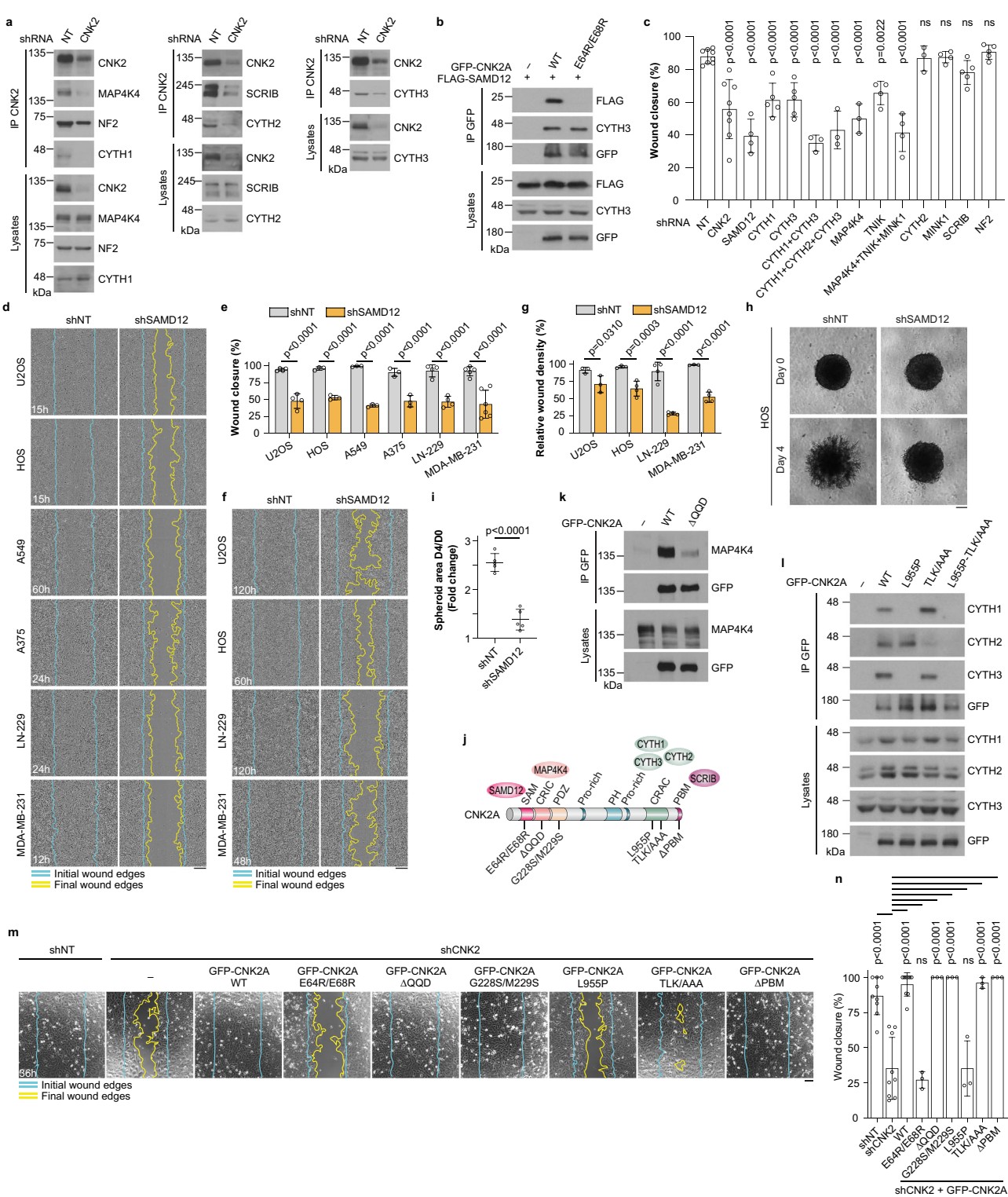

enriched for FLAG staining at the cell periphery (Supplementary Fig. 7g, h). Additionally, GFP-CNK2A WT and FLAG-CYTH1 colocalized strongly at sites of cell-cell contact, suggesting that CNK2A promotes the membrane localization of CYTH1. In agreement with this notion, compared to control cells, we detected less endogenous CYTH1 in the membrane fraction of CNK2-depleted cells (Fig. 5k). Overall, these results are consistent with the formation of a membrane-localized complex containing CNK2A, CYTH1, and ARF6.

To determine the mechanisms that regulate the presence of CNK2A at the plasma membrane, we used PIP strips[49] to identify phosphoinositide(s) that could specifically interact with a purified protein fragment encompassing the PH domain of CNK2A WT or W591A. While the CNK2A PH domain carrying the W591A mutation did not interact with any phosphoinositides, the WT PH domain interacted preferentially with PI(3,4,5)P3 and PI(3,4)P2, but also weakly with PI(4,5)P2 (Fig. 5l; Supplementary Fig. 7i, j). Since the former two phosphoinositides are catalyzed by PI3K[50], we inhibited PI3K using LY294002 to test its requirement for the membrane localization of endogenous CNK2. In addition to impairing cell migration and AKT activation (pAKT S473) (Supplementary Fig. 7k–m), LY294002

**Fig. 3 | CYTH1, CYTH3, and SAMD12 are functional CNK2 interactors required for cell migration. a** Endogenous CNK2 interacted with CYTH1,2,3, MAP4K4, NF2 and SCRIB by co-IP in U2OS cells. Immunoblots were probed with antibodies indicated on the right. **b** Co-IP experiments in U2OS cells indicated that FLAG-SAMD12 associates with GFP-CNK2A WT, but not with GFP-CNK2A_E64R/E68R. **c** U2OS cells were depleted of selected CNK2A BioID hits. Quantification of percent wound closure from wound healing assays after 15 h. For statistical analysis, each treatment group was compared to the control (shNT). **d** shRNA-mediated depletion of SAMD12 delayed migration of six cancer cell lines in wound healing assays. **e** Quantification of the migration assays shown in (**d**) at the indicated time points for each cell line. **f** Depletion of SAMD12 delayed invasion of the indicated cell lines used in Matrigel-based wound healing assays. **g** Quantification of the invasion assays shown in (**f**), at the indicated time points for each cell line. **h** Depletion of SAMD12 by shRNA reduced 3D invasion of HOS cell spheroids into Matrigel. **i** Quantification of the invasion assays shown in (**h**). **j** Schematic representation of

CNK2A with some of its interactors. Mutations tested in subsequent co-IP experiments and in functional migration assays are indicated. **k** Co-IP experiments in U2OS cells indicated that the QQD deletion (amino acids 172-174) in the CRIC region strongly disrupted binding with endogenous MAP4K4. **l** The L955P mutation in the CRAC region prevented association with CYTH1/3 and TLK (amino acids 961-963) into AAA substitution disrupted binding with CYTH2. **m** Expression of shRNA-insensitive GFP-CNK2A_E64R/E68R or GFP-CNK2A_L955P did not rescue the migration delay observed in CNK2-depleted U2OS cells in wound healing-like assays. **n** Quantification of the migration assays shown in **m**. All statistical analyses were performed on data from at least three independent experiments. One-way ANOVA was used in (**c**) and (**n**), two-way ANOVA was used in (**e**) and (**g**), and unpaired *t*-test was used in (**i**). Error bars correspond to mean values ± SD; ns, not significant ($P > 0.05$). Scale bar: **d**, **f** 200 μm, **h**, **m** 100 μm. Source data are provided as a Source Data file.

treatment decreased CNK2 membrane staining and its detection in the membrane fraction (Fig. 5m, n; Supplementary Fig. 7n). Thus, our data indicate that the role of CNK2 in promoting cell migration and ARF6 activation requires its ability to localize at the plasma membrane, which is mediated through binding of its PH domain with PI3K-dependent phospholipids.

## CNK2 couples AXL signalling to downstream ARF6 activation

Since CNK2 functions at the plasma membrane, we hypothesized that its localization could be regulated by growth factors and their receptors. Consistent with this idea, membrane staining of endogenous CNK2 in U2OS cells was strongly reduced by serum starvation (Fig. 6a; Supplementary Fig. 8a). Since CNK1 and *Drosophila* CNK were shown to function downstream of RTKs[5,15], we reasoned that this could also be the case for CNK2. To identify receptors that could regulate its presence at the membrane, we screened a panel of RTK inhibitors for their effect on CNK2 localization during cell migration (Fig. 6a; Supplementary Fig. 8a). Compared to cells treated with DMSO, treatment with two inhibitors that both target AXL (Cabozantinib and Bemcentinib) strongly decreased CNK2 membrane staining. Depletion of AXL with two different shRNAs corroborated this result (Fig. 6b–d), and demonstrated that reduced AXL expression in U2OS, HOS, and LN-229 cells delays migration (Supplementary Fig. 8b, c), like downregulation of CNK2. AXL, a member of the TAM (TYRO3, AXL, MER) RTK family, mediates actin cytoskeleton rearrangements and focal adhesion turnover to promote cell migration and invasion[51,52], and our results are thus consistent with its reported role in cell motility. Markedly, stimulation of serum-starved U2OS cells with the AXL ligand GAS6[51,52], which led to activation of AXL (pAXL Y702) and PI3K (pAKT S473), restored the membrane localization of CNK2 (Fig. 6e–g). This latter effect was abrogated by PI3K inhibition with LY294002 (Fig. 6e–g), suggesting that PI3K functions downstream of AXL to recruit CNK2 to the plasma membrane. Treatment with three other RTK inhibitors (Gefitinib, Crizotinib, and BMS-536924) also caused a statistically significant reduction of CNK2 membrane staining but their effect was much weaker than inhibition of AXL (Fig. 6a; Supplementary Fig. 8a).

Since GAS6-induced activation of AXL was sufficient to recruit CNK2 to the plasma membrane, we wanted to verify that GAS6 also stimulated cell migration and invasion. First, we used blocking antibodies against GAS6 or AXL to block receptor-ligand interactions and conducted wound healing migration and invasion assays. The addition of either antibody to the cell culture medium led to a decrease of AXL activation (reduced pY702 levels) and inhibited cell migration and invasion of U2OS cells (Supplementary Fig. 8d-h). Additionally, we tested the effect of GAS6 knockdown on U2OS and LN-229 cell migration and invasion using two different shRNAs. Like depletion of AXL, knockdown of GAS6 strongly reduced cell migration and invasion, indicating that secreted GAS6 stimulates motility in an autocrine and/or paracrine manner (Supplementary Fig. 8i–m). To test whether

extracellular GAS6 is sufficient to promote cell migration, we stimulated serum-starved U2OS cells with recombinant GAS6 and conducted wound healing migration assays. We found that this treatment promotes cell migration, and strikingly, that GAS6-induced migration is suppressed not only by knockdown of AXL but also by knockdown of CNK2, demonstrating that AXL and CNK2 are both required for GAS6-dependent migration in U2OS cells (Supplementary Fig. 9a–c). Finally, since CNK2 is regulated by AXL and is required for GAS6-dependent migration, we hypothesized that it may also be required for GAS6-induced single-cell chemotaxis. Using the transwell migration and invasion assay, we found that individual cell motility induced by 10% FBS and by recombinant GAS6 was strongly reduced by knockdown of CNK2, indicating that CNK2 is indeed required for chemotaxis toward growth factors including GAS6 (Fig. 6h, i; Supplementary Fig. 9d, e).

Next, since AXL regulates the localization of CNK2 and cell motility, we used pull-downs to test whether it also impacts ARF6 activity. Indeed, like CNK2 depletion, knockdown of AXL caused a reduction of ARF6-GTP levels in U2OS, LN-229, and A549 cells (Fig. 4a; Supplementary Fig. 9f, g). Strikingly, in U2OS cells, GAS6-induced activation of AXL triggered an increase of ARF6-GTP levels that was suppressed by CNK2 knockdown (Fig. 6j). In contrast, depletion of CNK2 had no effect on AXL-dependent pAKT, indicating that CNK2 is not required for PI3K activation (Fig. 6j). Therefore, these results show that AXL acts upstream of PI3K to recruit CNK2 to the plasma membrane and that CNK2 is required for AXL-dependent activation of ARF6. Taken together, our data support a model (Fig. 6k) in which extracellular pro-motility cues, including GAS6, activate the RTK AXL and trigger PI3K-dependent translocation of CNK2 to the plasma membrane. At the membrane, CNK2 mediates the activation of ARF6 through functional interactions with SAMD12, CYTH1, and CYTH3. Once activated, ARF6 signalling stimulates RAC1 and inhibits RHOA resulting in a positive effect on cell motility.

## CNK2 and SAMD12 promote metastasis in vivo

To assess the function of CNK2 and its binding partner SAMD12 in an in vivo context, we used CRISPR-Cas9 to knockout their respective genes in osteosarcoma cells and generated two clonal cell lines for each KO. We chose the HOS derivative 143B cell line as an osteosarcoma model because of their high tumorigenicity in mice[53]. Compared to control 143B cells (empty vector), *CNKSR2^KO* or *SAMD12^KO* cells (Supplementary Fig. 10a, b) displayed delayed wound healing migration and invasion, and reduced spheroid invasion (Fig. 7a–f) recapitulating the effects of shRNA-mediated depletion of each protein in U2OS and HOS cells. Additionally, neither knockout affected cell viability (Supplementary Fig. 10c). To determine whether CNK2 and SAMD12 are involved in tumour growth and metastasis, we injected luciferase-expressing 143B control (empty vector), *CNKSR2^KO*, or *SAMD12^KO* cells intravenously through the tail vein of immunodeficient mice and monitored tumour development through bioluminescence

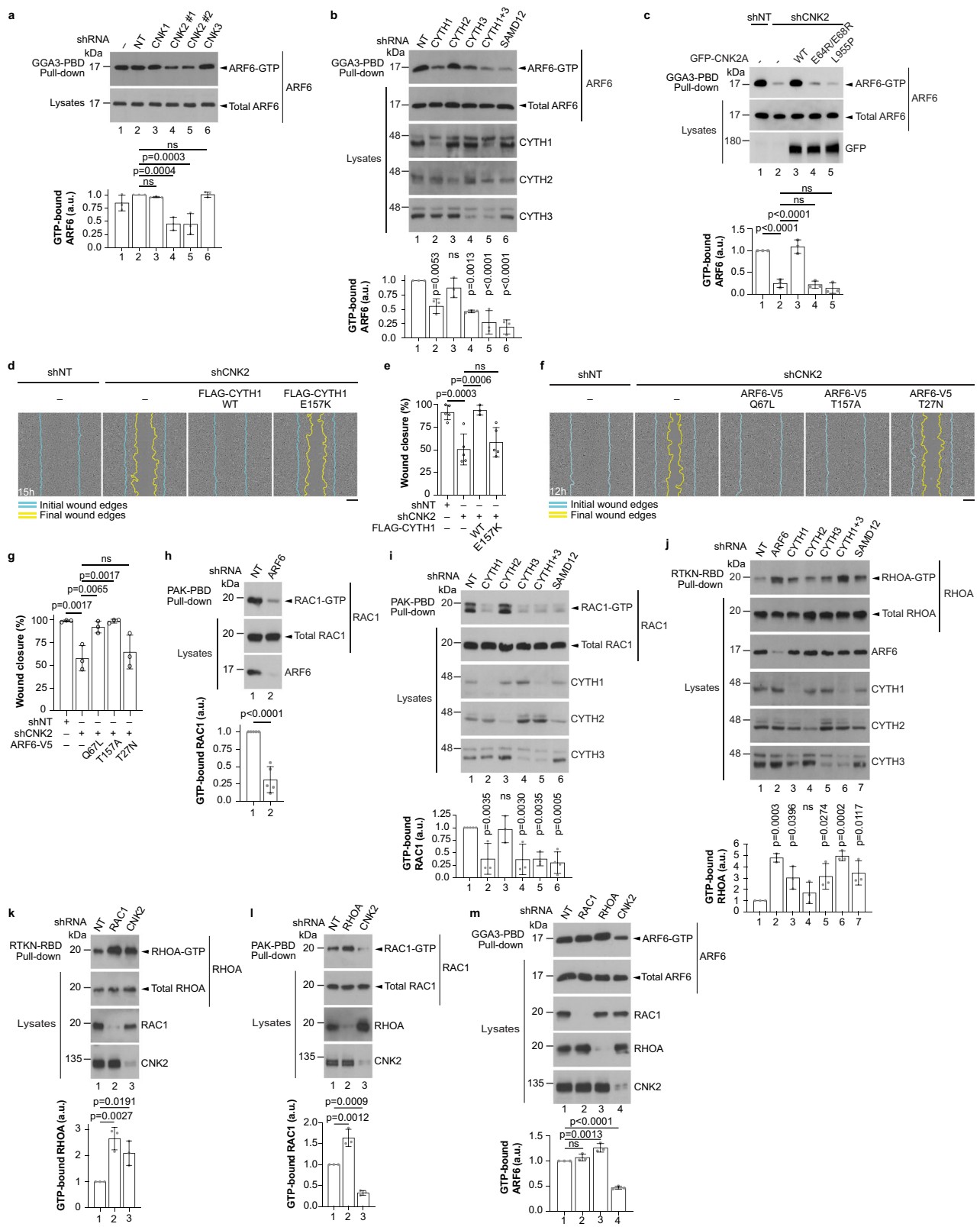

imaging at weekly intervals for 3 weeks. Mice injected with control cells displayed a strong bioluminescent signal in the abdominal region after 3 weeks, indicative of metastatic tumour formation, that was drastically reduced by both *CNKSR2* KO or *SAMD12* KO (Fig. 7g, h). Since the livers and lungs are common sites of metastasis following tail vein injection[54], we harvested these organs 3 weeks after injection and assessed the number of metastatic lesions (Supplementary Fig. 10d).

Compared to control cells, *CNKSR2*[KO] or *SAMD12*[KO] cells produced fewer whole-liver surface tumours and histological analysis of left lobes revealed significantly less internal lesions (Fig. 7i–k). KO of *CNKSR2* or *SAMD12* did not affect the size of liver tumours and liver weight (Supplementary Fig. 10e, f), suggesting that reduced tumour burden caused by loss of CNK2 or SAMD12 is due to impaired cell invasion and not tumour growth after colonization. Similarly to

**Fig. 4 | CNK2 promotes ARF6 activity upstream of RAC1 and RHOA. a–c** Binding of CNK2A with CYTH1, CYTH3 and SAMD12 promotes ARF6 activity. ARF6-GTP was precipitated using GGA3 protein-binding domain (PBD) beads. U2OS cells were depleted of endogenous CNKs, CYTHs or SAMD12 by shRNA and shRNA-insensitive GFP-CNK2A WT or mutant constructs were expressed where indicated. Bar graphs indicate the normalized ratio of GTP-ARF6 to the amount of total ARF6. **d** Overexpression of FLAG-CYTH1 WT, but not FLAG-CYTH1_E157K, rescued the migration delay observed in CNK2-depleted U2OS cells in wound healing assays. **e** Quantification of the migration assays shown in (**d**). **f** Expression of constitutively active (Q67L) or fast cycling (T157A) ARF6-V5, but not the dominant negative

mutant (T27N), rescued the migration delay observed in CNK2-depleted U2OS cells in wound healing assays. **g** Quantification of the migration assays shown in (**f**). **h**–**m** GTP-bound RAC1 (**h**, **i**, **l**), RHOA (**j**, **k**), and ARF6 (**m**) were precipitated using PAK-PBD, Rhotekin-RBD or GGA3-PBD beads as indicated. Migrating U2OS cells were depleted of the indicated proteins by shRNA. Bar graphs indicate the normalized ratio of GTP-bound GTPase to the amount of total proteins. All statistical analyses were one-way ANOVA and were performed on data from at least three independent experiments. For **b**, **i**, and **j** each treatment group was compared to the control (shNT). Error bars correspond to mean values ± SD; ns, not significant ($P > 0.05$). Scale bar: **d**, **f** 200 μm. Source data are provided as a Source Data file.

livers, histology revealed that injection of *CNKSR2^(KO)* or *SAMD12^(KO)* cells led to significantly fewer internal metastatic lesions in the lungs compared to control cells (Fig. 7i, l). Overall, these data indicate that CNK2 and SAMD12 are required for metastasis in an in vivo cancer model.

## Discussion

In this study we discovered a CNK2-dependent signalling axis triggered by the RTK AXL that promotes cancer cell migration, invasion, and metastasis. We elucidated a crucial function of CNK2 in stimulating the activation of the small GTPase ARF6, a key regulator of membrane trafficking and cytoskeletal remodelling. Regulation of ARF6 is mediated by the CNK2A isoform (Fig. 1a), which contains the C-terminal CRAC region, also termed the cytohesin-binding domain[15]. Our data support a model in which the scaffold CNK2A forms a complex with CYTH1 and CYTH3 at the plasma membrane and acts as a platform to facilitate ARF6 activation (Fig. 6k). The CNK2 homologs CNK1 and IPCEF1 (Fig. 1a) also interact with cytohesins (CYTH1, 2, or 3) and were shown to regulate ARF GTPases[15,16]. Notably, CNK1 promotes insulin-dependent activation of ARF1 and ARF6 in hepatocellular carcinoma cells[15], while IPCEF1 overexpression enhances ARF6 activity in EGF-stimulated COS cells[16]. These functions of CNK1 and IPCEF1 involve binding with CYTH2, but it is unclear whether CYTH1 or CYTH3 are involved. Intriguingly, our study reveals that the CNK2A binding interface differs between CYTH1/CYTH3 and CYTH2 and that binding with CYTH1 and CYTH3, but not CYTH2, is crucial for CNK2A-dependent activation of ARF6 during cell migration. Our results also suggest at least a partial redundancy between CYTH1 and CYTH3 in regulating cell migration. However, it remains unclear how CNK2A binding with CYTH1 and CYTH3 promotes nucleotide exchange on ARF6. It is possible that this interaction enhances cytohesin GEF activity through an allosteric mechanism, or that another CNK2A binding partner regulates cytohesin GEF activity. Conversely, CYTH1 and CYTH3 could be activated independently of CNK2A, and the function of CNK2A may be limited to recruiting them to the plasma membrane in proximity with ARF6. Nevertheless, our findings identify a mechanism that differentially regulates the activity of cytohesins toward ARF6 and suggest that CNK2A might function in distinct ARF-dependent signalling pathways.

The function of CNK2A in migration and ARF6 activation also requires binding with the SAM domain-containing protein SAMD12, which interacts with the CNK2A SAM domain. SAMD12 is an ortholog of the *Drosophila* CNK interactor Hyphen, which mediates the formation of a complex containing the pseudokinase KSR and the kinase RAF. This complex promotes RAF activation and RAS/MAPK signalling[34,35], however, the role of SAMD12 in this pathway in mammals is unknown. The molecular function of SAMD12 is still unclear, but interestingly, like *CNKSR2* mutations, a repetitive insertion mutation in the *SAMD12* locus has been associated with myoclonic epilepsy[55,56], indicating that it has a neuronal function like CNK2. While the mechanism by which SAMD12 promotes CNK2A-dependent ARF6 activity remains to be identified, as a putative adaptor protein it might act allosterically to stabilize a CNK2A-cytohesin complex that favours nucleotide exchange on ARF6. SAMD12 could also function by

mediating binding between CNK2 and another interactor(s) that is necessary for cytohesin-dependent or -independent ARF6 activation. More experiments are required to decipher the functional interplay between cytohesin and SAMD12 binding to CNK2A and how this complex activates ARF6.

We found that CNK2 promotes cell migration by coordinating the activity of RAC1 and RHOA, two crucial regulators of cell morphology and motility. CNK2 was previously shown to regulate RAC1 during spine morphogenesis in rat hippocampal neurons through binding with the RacGAP Vilse[17]. In contrast, here we show that CNK2 promotes RAC1 activity, and inhibits RHOA, through a different mechanism involving cytohesin-dependent ARF6 activation. This suggests that a fundamental role of CNK2 is to control RHO GTPase signalling and that this can occur through cell type-dependent mechanisms. While it is currently unclear how ARF6 regulates RAC1 and RHOA downstream of CNK2 to promote cell migration, several mechanisms of ARF6-dependent RAC1 activation have been described. In MDCK cells, ARF6 promotes RAC1 activity by recruiting the RAC GEF DOCK180 (also named DOCK1) to the plasma membrane[46], and interestingly, the CNK family member IPCEF1 appears to function in this process by binding both CYTH2 and DOCK180 (ref. 57). These data raise the possibility that CNK2A enables positive ARF6-to-RAC1 crosstalk by simultaneously scaffolding ARF GEFs and RAC GEFs, although no RAC GEFs have been recovered in our BioID screens. ARF6 also promotes RAC1 activation through its role in membrane trafficking. In HeLa cells, ARF6 controls the recycling of RAC1-containing endosomes required for membrane ruffling[47,58] and in mouse embryonic fibroblasts ARF6 mediates trafficking of lipid rafts that contribute to RAC1-dependent cell spreading[59]. Interestingly, ARF6 has also been shown to induce the turnover of adherens junctions[60,61] and focal adhesions[62], which are important sites of RHO GTPase activity[63–65]. Since we found that CNK2 localizes at adherens junctions and limits the maturation of focal adhesions, ARF6-dependent trafficking to these sites could serve as a mechanism that regulates RAC1 and RHOA. Although inhibition of RHOA by ARF6 is poorly understood, our data are consistent with previous reports that RAC1 and RHOA function in an antagonistic manner[33], suggesting that ARF6 could regulate RAC1 upstream of RHOA. In this case, antagonism of RHOA could be achieved by RAC1-mediated inhibition of RHOA GEFs or stimulation of its GAPs[33]. However, the identification of ARF6-regulated enzymes directly inhibiting RHOA is also an appealing prospect. Interestingly, there are several ARF GAPs that also contain RHO-GAP domains, namely, ARAP1, ARAP2, and ARAP3 (ref. 66). As some ARF GAPs also appear to work as ARF effectors[66], it is possible that one or more ARAPs serve this function on ARF6 towards RHOA. More work is necessary to identify the downstream functions of ARF6 that are regulated by CNK2A and how they impact the spatiotemporal dynamics of RAC1 and RHOA signalling during cell motility.

Our findings indicate that CNK2 functions downstream of GAS6-induced AXL signalling. This RTK is overexpressed in many cancers and confers motile and invasive phenotypes, yet the signalling mechanisms mediating its function are only recently being uncovered[67]. In breast cancer cells, AXL promotes RAC1-dependent invasion through phosphorylation of ELMO[68], a scaffold for DOCK

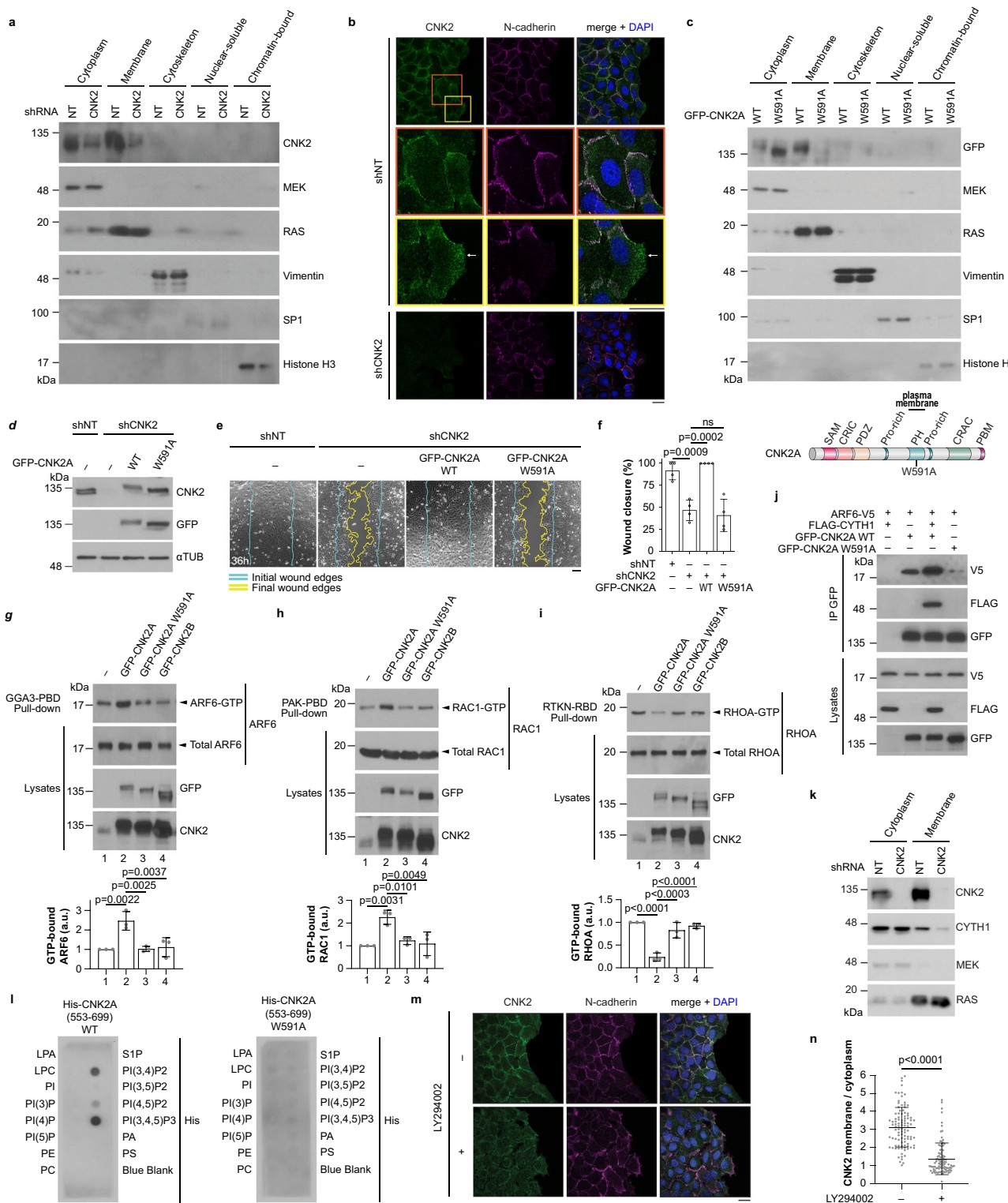

family RAC GEFs, while in glioblastoma cells AXL-dependent invasion involves PI3K and RAC1 activity[52]. AXL also regulates breast cancer cell motility by triggering a kinase cascade that culminates in CSK-mediated phosphorylation of paxillin, a signal that is required for focal adhesion disassembly and turnover[51]. Strikingly, our study uncovers CNK2-dependent ARF6 signalling as a novel mechanism underlying the pro-motility function of AXL in U2OS osteosarcoma cells. Since we observed that AXL and CNK2 also stimulate ARF6 activity in LN-229 (glioblastoma) and A549 (lung adenocarcinoma) cells, this strongly suggests that the AXL-CNK2-ARF6 axis is conserved

across multiple cancer cell types. However, considering that CNK2 function is also dependent on PI3K and that the activity of many cell surface receptors converges on PI3K activation[69], it will be important to determine which other extracellular cues and receptors can regulate CNK2A in cancer cells.

Intriguingly, despite its prominent neuronal expression in healthy tissues[7] (Supplementary Fig. 11), we found that CNK2A promotes cell motility in several non-neuronal cancer cell lines. Our findings add to emerging reports of deregulated neuronal gene expression enhancing the motile phenotype of non-neuronal cancer cells[70,71]. While

**Fig. 5 | CNK2 functions at the plasma membrane. a** Fractionation of U2OS cell lysates followed by immunoblotting indicated that endogenous CNK2 is detected in the cytoplasmic and membrane fractions. shRNA-mediated depletion of CNK2 confirmed the specificity of our custom-made anti-CNK2 antibody. Compartment-specific markers were probed to validate the different fractions. These markers included: MEK (cytosolic fraction), RAS (membrane fraction), Vimentin (cytoskeletal fraction), SP1 (nuclear-soluble fraction) and Histone H3 (chromatin-bound fraction). **b** Endogenous CNK2 colocalized with N-cadherin in migrating U2OS cells. Note that CNK2 is also observed at the leading edge of some migrating cells (arrow). Orange and yellow boxes in the upper left image are magnified below. CNK2 was depleted by shRNA to control for signal specificity. **c** Fractionation experiments of U2OS cell lysates expressing GFP-CNK2A WT or GFP-CNK2A_W591A indicated that mutation in the PH domain abolished the detection of GFP-CNK2A in the membrane fraction. A schematic representation of CNK2A with the PH domain mutation (W591A) is shown below. **d** Immunoblots from U2OS cells depleted of CNK2 and expressing shRNA-insensitive GFP-CNK2A WT or GFP-CNK2A_W591A. **e** Expression of shRNA-insensitive GFP-CNK2A WT but not GFP-CNK2A_W591A rescued the migration delay observed in CNK2-depleted U2OS cells in wound healing-like assays. **f** Quantification of the migration assays shown in (**e**). **g–i** Overexpression of GFP-CNK2A WT in migrating U2OS cells increased ARF6-GTP and RAC1-GTP levels

and decreased RHOA-GTP levels while overexpression of GFP-CNK2A_W591A or GFP-CNK2B had no effect on GTPase activity. GTP-bound ARF6 (**g**), RAC1 (**h**), and RHOA (**i**) were precipitated using GGA3-PBD or PAK-PBD or Rhotekin-RBD beads, respectively. Bar graphs indicate the normalized ratio of GTP-bound GTPase to the amount of total proteins. **j** ARF6-V5 co-immunoprecipitated with GFP-CNK2A but not with GFP-CNK2A_W591A in HEK293T cells. The ARF6-V5 interaction with GFP-CNK2A was enhanced by co-expression with FLAG-CYTH1. **k** Fractionation of U2OS cell lysates showing reduced CYTH1 levels in the membrane fraction following CNK2 depletion. **l** Purified truncated versions of recombinant His-CNK2A, consisting of residues 553-699 (WT or W591A) and containing the PH domain, were used to probe PIP Strips for lipid binding. Bound proteins were detected using anti-His antibodies. **m, n** Inhibition of PI3K decreased CNK2 localization at the plasma membrane in migrating U2OS cells. Cells were treated with LY294002 (10 μM) for 16 h and stained for endogenous CNK2 and N-cadherin. 100 cells were quantified for each condition. All statistical analyses were performed on data from three independent experiments. One-way ANOVA was used in (**f–i**), and unpaired $t$-test was used in **n**. Error bars correspond to mean values ± SD; ns, not significant ($P > 0.05$). Scale bar: **b, m** 20 μm, **e** 100 μm. Source data are provided as a Source Data file.

further work is required to determine if, and how, the *CNKSR2* locus is derepressed in non-neuronal cancer cells, our study suggests that aberrant expression of CNK2A confers a motile phenotype by hijacking ARF6 signalling. Such a prospect could make CNK2A a relevant biomarker for metastatic cancers.

In summary, our work identifies the scaffold protein CNK2A and its binding partner SAMD12 as crucial regulators of cancer cell motility and invasion. Targeting their function with small molecules might provide new therapeutic means for preventing metastasis. Given the number and diversity of common CNK2 and CNK3 interactors, it is likely that a detailed understanding of these protein networks will uncover novel mechanisms underlying cell morphogenesis in normal and pathophysiological conditions.

## Methods

All experiments involving mice were approved by the Université de Montréal Institutional Animal Care Committee in compliance with guidelines from the Canadian Council on Animal Care (CCAC).

### Cell culture

U2OS cells were originally purchased from ATCC (HTB-96™) and obtained from Katherine L. B. Borden's lab (IRIC, Université de Montréal), and were grown in Dulbecco's modified Eagle's medium (DMEM, Sigma-Aldrich) supplemented with 5% fetal bovine serum (FBS, Wisent). LN-229 (ATCC® CRL-2611™), MDA-MB-231 (ATCC® HTB-26™), HOS (ATCC® CRL-1543™) and 143B (ATCC® CRL-8303™) cells were maintained in DMEM supplemented with 10% FBS. HEK293T (Sigma-Aldrich; #12022001-1VL) were grown in DMEM supplemented with 5% FBS. A375 (ATCC® CRL-1619™) and A549 (ATCC® CCL-185™) cells were cultured in RPMI-1640 medium (Gibco) supplemented with 10% FBS. Flp-In T-REx HEK293 and Flp-In T-REx U2OS cells were kindly provided by Anne-Claude Gingras (University of Toronto, Canada) and Stephen Blacklow (Harvard Medical School, USA), respectively. All cell lines derived from Flp-In T-REx cells and used in BioID experiments were cultured in DMEM supplemented with 5% tetracycline-free FBS. All the above cell lines were cultured at 37 °C and 5% $CO_2$. hFOB 1.19 cells (ATCC® CRL-11372™) were maintained in DMEM/Nutrient Mixture F-12 Ham culture medium (Sigma-Aldrich; #D6434) containing 2.5 mM L-glutamine (Sigma-Aldrich; #G7513) and 10% FBS at 34 °C and 5% $CO_2$. The culture media and source for the 39 cell lines used in RT-qPCR experiments and referred to in "Results: CNK2 promotes cancer cell migration and invasion" are shown in Supplementary Data 2. All cell lines used in this study were routinely tested by PCR for mycoplasma contamination.

### Plasmids

Plasmids expressing shRNA-insensitive GFP-CNK2A WT and shRNA-insensitive GFP-CNK2B were generated by Gateway recombination (Invitrogen). Coding sequences were first cloned into the pDONR221 entry vector and then recombined into the pLVpuro-CMV-N-EGFP destination vector (Addgene; #122848) used for lentivirus production and transduction in mammalian cells. The following expression vectors were generated: pLV-GFP-CNK2A WT, pLV-GFP-CNK2B WT, pLV-GFP-CNK2A_E64R/E68R, pLV-GFP-CNK2A_ΔQQD, pLV-GFP-CNK2A_G228S/M229S, pLV-GFP-CNK2A_W591A, pLV-GFP-CNK2A_L955P, pLV-GFP-CNK2A_TLK/AAA and pLV-GFP-CNK2A_ΔPBM.

The cDNA sequence coding for human SAMD12 (amino acids 1-201) was cloned into pDONR221 and recombined into pLVpuro-CMV-N-3xFLAG (Addgene; #123223) to generate pLV-FLAG-SAMD12.

The pDONR221 containing the coding sequence of human CYTH1 was purchased from Addgene (#117447; g4s pDONR) and was recombined into the pLVpuro-CMV-N-3xFLAG vector (Addgene; #123223). We generated two CYTH1 plasmids: pLV-FLAG-CYTH1 WT and pLV-FLAG-CYTH1_E157K.

The pDONR221 containing the coding sequence of human ARF6 was purchased from Addgene (#67314; pDONR221-ARF6). In this plasmid, the V5 tag sequence, followed by a stop codon, was added at the C-terminus of ARF6 to obtain a pDONR221 ARF6 WT-V5 vector. The ARF6 WT-V5 sequence was then recombined into the doxycycline inducible lentiviral pCW57.1 vector (Addgene; #41393) using Gateway cloning. The ARF6 Q67L, T27N and T157A mutations were generated by site directed mutagenesis in the pCW57.1 ARF6 WT-V5 vector.

pOG44 Flp-Recombinase Expression Vector (Invitrogen; #V600520) was a gift from Dr. Jean-François Côté (IRCM, Montreal). All constructs used to generate stable cell lines in Flp-In T-REx HEK293 and Flp-In T-REx U2OS cells were cloned in pcDNA5 FRT/TO FLAG-BirA R118G (BirA*) and obtained from the laboratory of Philippe Roux (IRIC, Montreal). The following expression vectors were generated: pcDNA5 FLAG-BirA*, pcDNA5 FLAG-BirA*-EGFP, pcDNA5 FLAG-BirA*-CNK1, pcDNA5 FLAG-BirA*-CNK2A, pcDNA5 FLAG-BirA*-CNK3.

For bacterial expression, the cDNA sequence of human CNK2A was codon optimized in the pUC57 plasmid (GenScript, USA). Sequences coding for CNK2A[1–309] and CNK2A[553–699], were then cloned in pPROEX with an N-terminal TEV protease-cleavable 6x His tag.

All plasmids used for lentivirus and retrovirus production and CRISPR/Cas9 are described in the "Lentiviral and retroviral infections" and "Generation of CRISPR/Cas9 knockout cells" sections, respectively.

For the bioluminescence experiments in mice, we purchased the MSCV Luciferase PGK-hygro plasmid from Addgene (#18782).

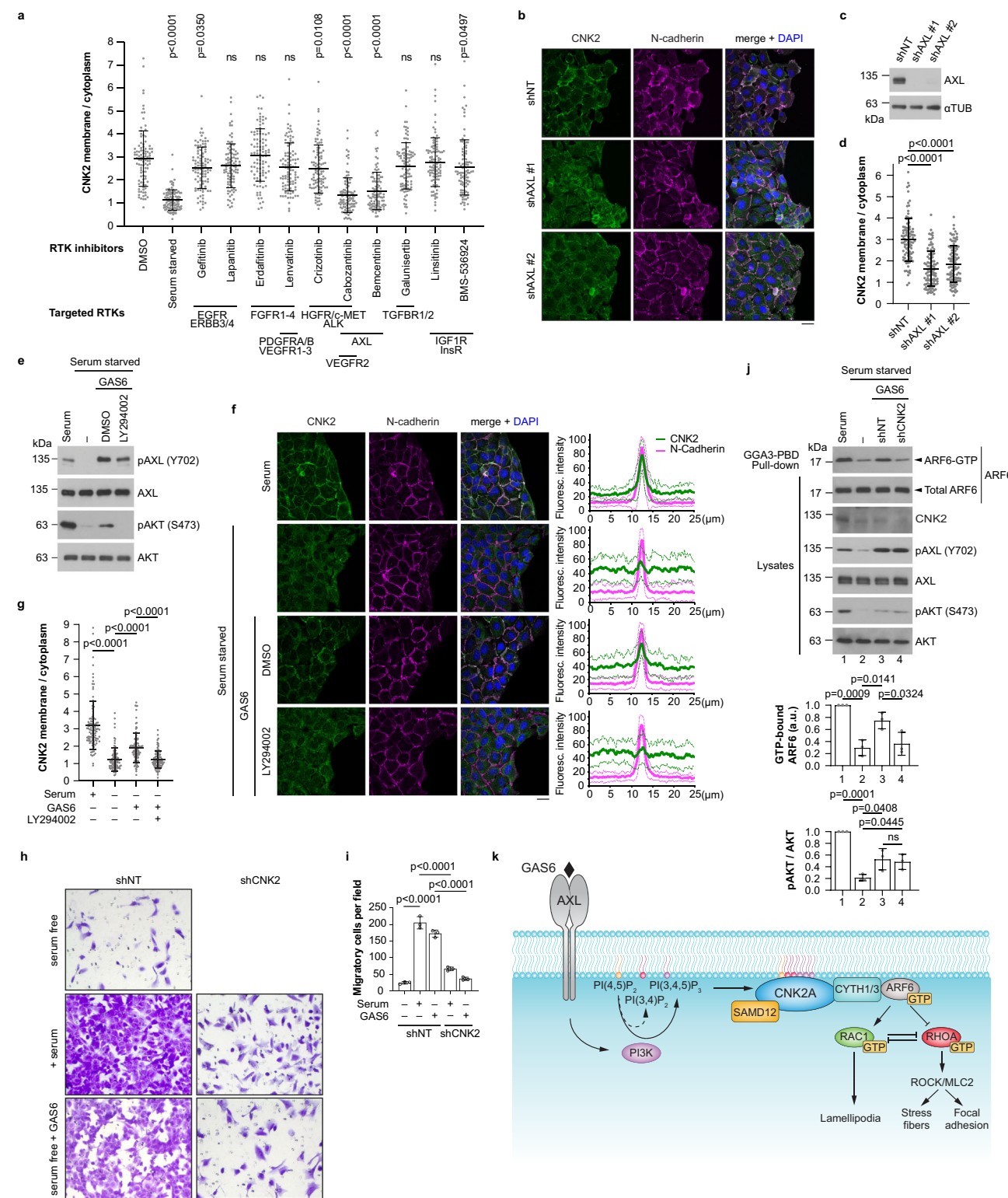

All mutations were generated using QuikChange II XL Site-Directed Mutagenesis Kit (Agilent; #200522) following the manufacturer's protocol. All constructs were verified by Sanger sequencing.

**Proximity-dependent biotin identification (BioID)**
Stable cell lines expressing FLAG-BirA*, FLAG-BirA*-EGFP, FLAG-BirA*-CNK1, FLAG-BirA*-CNK2A, and FLAG-BirA*-CNK3 were generated by co-transfection of pOG44 Flp-Recombinase and pcDNA5 FLAG-BirA* fusion constructs in Flp-In T-REx cells with lipofectamine 2000 and

were selected with 200 μg/mL Hygromycin B. The stable cell lines were maintained in DMEM supplemented with 5% tetracycline-free FBS and 200 μg/mL Hygromycin B.

For affinity purification, Flp-In T-REx HEK293 and Flp-In T-REx U2OS stable cell lines expressing the fusion proteins of interest were induced with 1 μg/mL tetracycline (Fisher Scientific; #BP912) and 50 μM D-biotin was added for 24 hours to label proteins (BioBasic; #BB0078). Cells were then washed twice with cold PBS (Millipore Sigma; #D8537), spun at 100 × *g* for 5 min at 4 °C and cell pellets were

**Fig. 6 | CNK2 couples AXL signalling to downstream ARF6 activation. a** Plasma membrane localization of CNK2 was strongly reduced by serum starvation or inhibition of the AXL receptor tyrosine kinase (RTK). Migrating U2OS cells were either serum starved or treated with 1 μM of the indicated RTK inhibitors. For statistical analysis, each treatment group was compared to the control (DMSO). **b**–**d** Depletion of AXL in U2OS cells with two different shRNAs strongly decreased CNK2 localization at the plasma membrane. **e**–**g** Stimulation of serum-starved U2OS cells with 200 ng/mL GAS6, for 30 min, increased phosphorylation of AXL (Y702) and AKT (S473) (**e**), and triggered PI3K-dependent recruitment of CNK2 to the plasma membrane (**f, g**). N-cadherin staining was used as a marker of the plasma membrane. Line scans correspond to the average ± SD immunofluorescence signal intensity for CNK2 in the cytoplasm and at the plasma membrane of two adjacent cells (*n* = 30 line scans for each condition). Bold lines indicate the average signal, while the thinner lines represent the standard deviation. All signals were normalized to the strongest signal for each channel. **h** Migration of U2OS cells towards 10% FBS and 200 ng/mL GAS6 in the transwell assay was strongly reduced by CNK2 depletion. **i** Quantification of the average number of cells per field for the transwell migration assay shown in (**h**). **j** Depletion of CNK2 suppressed GAS6-dependent activation of ARF6. Migrating U2OS cells were serum starved for 24 hours and stimulated with 200 ng/mL GAS6 for 30 min. Bar graphs indicate the normalized ratio of GTP-ARF6 to total ARF6 or pAKT (S473) to total AKT. **k** Proposed model. The scaffold protein CNK2 promotes cancer cell motility by mediating ARF6 activation downstream of AXL signalling (see details in the text). All statistical analyses were one-way ANOVA and were performed on data from three independent experiments. In **a, d, g**, 100 cells were quantified for each condition. Error bars correspond to mean values ± SD; ns, not significant (*P* > 0.05). Scale bar: **b, f** 20 μm, **h** 200 μm. Source data are provided as a Source Data file.

frozen on dry ice. Cell pellets were lysed in RIPA buffer (1% of IGEPAL (Millipore Sigma; #I3021), 0.1% SDS (Fisher Scientific; #BP166), 50 mM Tris-HCl pH7.5 (BioShop; #TRS001), 150 mM NaCl (Millipore Sigma; #SX0420-5), 12.7 mM sodium deoxycholate (Millipore Sigma; #D6750) and 1 mM EDTA (Fisher Scientific; #BP120)) supplemented with 167 units of Benzonase (Millipore Sigma; #E1014) on ice for 20 min, sonicated (20% amplitude) and spun at $19,000 \times g$ for 20 min at 4 °C. Cleared lysates were incubated with 15 μL of High Performance Streptavidin Sepharose beads (Fisher Scientific; #45-000-279) for 3 h at 4 °C and washed 3 times with RIPA lysis buffer and 3 times with 50 mM ammonium bicarbonate (Millipore Sigma; #A6141). Beads were resuspended in 100 μL 50 mM of ammonium bicarbonate, and trypsin digestion was performed overnight at 37 °C by adding 1 μg of trypsin (Millipore Sigma; #T6567) to the beads. The following day, an additional 1 μg of trypsin was added for 4 h and beads were centrifuged. Supernatants were then transferred into new microcentrifuge tubes, the beads were washed twice with 100 μL HPLC H$_2$O, and each wash was pooled with the supernatant (Fisher Scientific; #W5-4). To stop digestion, 4% formic acid (Millipore Sigma; #94318) was added to the samples. Samples were then spun at $11,000 \times g$ for 10 min, transferred into new microcentrifuge tubes, dried in a SpeedVac and solubilized in 5% acetonitrile (Fisher Scientific; #ACN; A998-4) and 0.2% formic acid (Fisher Scientific; #AC147930010).

## Mass Spectrometry analysis

Samples purified from Flp-In T-REx HEK293 cells were reduced, alkylated with chloroacetamide and digested with trypsin. Tryptic digests were separated on a homemade reversed-phase column (150-μm i.d. by 200 mm) with a 56-min gradient from 10 to 30% acetonitrile, 0.2% formic acid and a 600 nL/min flow rate on an Easy nLC-1200 connected to a Q-Exactive HF Biopharma (Thermo Fisher Scientific, San Jose, CA). Each full MS spectrum acquired at a resolution of 60,000 was followed by tandem-MS (MS-MS) spectra acquisition of the 15 most abundant precursors. Tandem-MS experiments were performed using higher energy collisional dissociation (HCD) at a normalized collision energy of 27%. Samples from Flp-In T-REx U2OS cells were treated the same as those from Flp-In T-REx HEK293 cells except that mass spectrometry analyses were performed on an Orbitrap Fusion (Thermo Fisher Scientific, San Jose, CA). Each full MS spectrum acquired at a resolution of 60,000 was followed by tandem-MS (MS-MS) spectrum acquisition on the most abundant multiply charged precursor ions for a maximum of 3 s. Tandem-MS experiments were performed using collision-induced dissociation (CID) at a collision energy of 30%. All data were processed using PEAKS X Pro (Bioinformatics Solutions, Waterloo, ON) and the Uniprot database (20,366 entries; Release 2020_03). Mass tolerances on precursor and fragment ions were 10 ppm and 0.01 Da, respectively, for the Flp-In T-REx HEK293 datasets and 10 ppm and 0.3 Da for Flp-In T-REx U2OS datasets. Fixed modification was carbamidomethyl (C). Variable selected posttranslational modifications were oxidation (M), deamidation (NQ), phosphorylation (STY). All data were visualized and exported with Scaffold Viewer 5.1.2.

## Interaction scoring

Datasets from at least 3 biological replicates with their respective negative controls consisting of affinity purifications from FLAG-BirA* and/or FLAG-BirA*-EGFP were merged and exported from Scaffold Viewer (protein threshold was set at 1.0% false-discovery rate (FDR) with ≥2 peptides identified and 1.0% FDR for peptides). To score the confidence of the interaction, we used SAINTexpress interaction scoring analysis from the Resource for Evaluation of Protein Interaction Networks (REPRINT) online platform (https://reprint-apms.org/). Proximal interactions with a SAINTexpress score ≥ 0.8 were represented using the Dot plot Generator from the ProHits-viz online platform (https://prohits-viz.lunenfeld.ca/) using the average spectral counts and the SAINTexpress score as the abundance and score parameters, respectively. Proximal interactors of interest were grouped using the gene ontology enrichment scoring from the STRING (version 11.5) functional protein association networks online platform (https://string-db.org/). For GO term analysis, the biological processes indicated in the dot plots had a false discovery rate (FDR) < 0.05. Network representation of CNK2 proximal interactors in U2OS cells was also generated using STRING.

## Lentiviral and retroviral infections

Viral particles were produced by first seeding three million HEK293T cells in T75 cell culture flasks. For lentiviruses, cells were transfected the next day with 5 μg of pLKO.1-shRNA (Millipore Sigma), 6 μg of pΔ8.9 (gag, pol) and 700 ng of pCMV-VSVG. For retroviruses, cells were transfected the next day with 6 μg of pCL-Ampho Retrovirus packaging vector and 6 μg of MSCV Luciferase PG-K-hygro. All transfections were done using polyethylenimine (PEI; 25 μg/mL) (Polysciences Inc. #23966). Viral supernatants were collected 72 h after transfection, filtered through 0.45 μm polyethersulfone (PES) filters (Sarstedt; #83.1826), aliquoted, and stored at −80 °C. For infections, cells were seeded at 30% confluence and transduction was facilitated by addition of polybrene (4 μg/mL). For shRNA experiments, puromycin (2.5 μg/mL) selection was applied 24 h post-infection and maintained for the duration of each experiment. For the generation of luciferase-expressing 143B cell lines, hygromycin B (500 μg/mL) selection was applied for a period of 5 days. All shRNA expression constructs were in the pLKO.1-puro backbone and sequences are listed in Supplementary Data 4 in the "shRNA sequences" tab.

## Cell lysates, immunoprecipitations, and western blotting

Co-immunoprecipitation experiments and western blotting procedures were essentially conducted as follows. All cell lines used in this study, except for HEK293T cells, were first washed once with cold PBS, then scraped in 1 mL cold PBS and were centrifuged at $100 \times g$ at 4 °C for 5 min. Cells were then lysed on ice for 20 min with lysis buffer

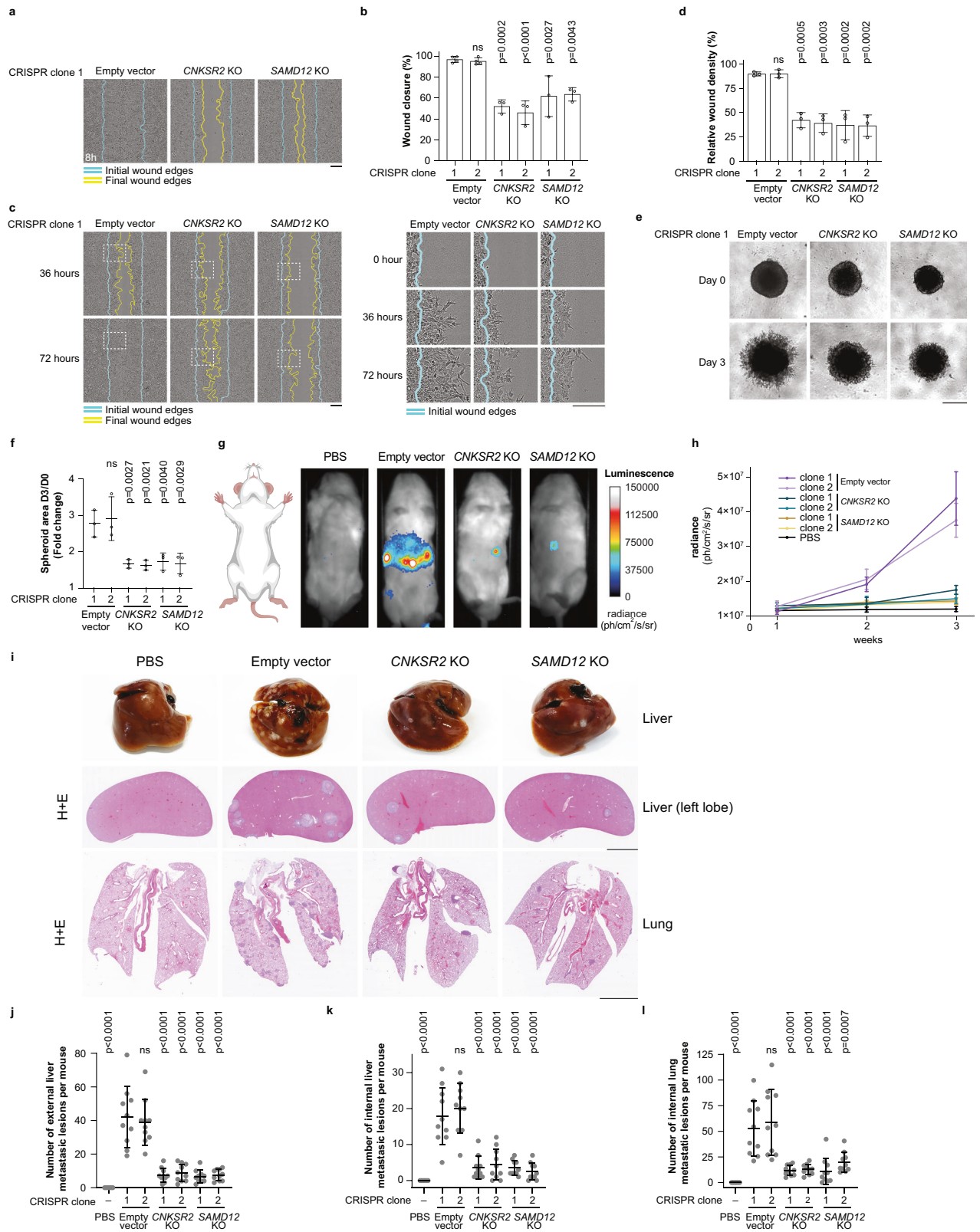

(50 mM Tris-HCl pH 7.5, 150 mM NaCl, 0.2-0.5% Triton X-100, 10% glycerol, 1 mM EDTA) containing 20 μM leupeptin, aprotinin (0.15 U/mL), 1 mM PMSF, phosphatase inhibitor cocktail (Sigma-Aldrich), and 1 mM sodium orthovanadate. HEK293T cells were washed once with cold PBS and then directly lysed in the dish for 20 min at 4 °C with gentle rocking. Cell lysates were centrifuged at 22,000 × g at 4 °C for 10 min and transferred into new microcentrifuge tubes.

For endogenous immunoprecipitation of CNK2, U2OS cell lysates were incubated with 5 μl of anti-CNK2 for 2 h and 10 μl of protein A/G agarose beads (Millipore Sigma) for 1 h, on a rotator at 4 °C. For co-IP experiments in HEK293T cells, 5×10⁶ cells were seeded overnight in 100 mm tissue culture plates, transfected the next day with the appropriate constructs using PEI and lysed 24 hours later. For immunoprecipitation of exogenous proteins, cell lysates were incubated

**Fig. 7 | CNK2 and SAMD12 promote metastasis in vivo. a** CRISPR/Cas9-mediated knockout (KO) of *CNKSR2* or *SAMD12* loci in 143B osteosarcoma cells delayed migration in wound healing assays. **b** Quantification of the migration assays shown in (**a**). Two clones were generated for each CRISPR KO and for control cells (empty vector). **c** *CNKSR2* and *SAMD12* KO in 143B cells delayed invasion in Matrigel-based wound healing assays. Magnified images of the dotted area at 0 h, 36 h and 72 h are shown on the right. **d** Quantification of the migration assays shown in (**c**). **e** KO of *CNKSR2* or *SAMD12* reduced 3D invasion of 143B cell spheroids into Matrigel. **f** Quantification of the invasion assays shown in (**e**). **g** CNK2 and SAMD12 regulate metastasis of 143B cells in vivo. Representative bioluminescent images of mice injected in the tail vein with PBS or 143B-Luciferase control, *CNKSR2^{KO}* or *SAMD12^{KO}* cells after 3 weeks. **h** Bioluminescence quantification of tumour growth at different weeks of mice injected with PBS (*n* = 2) and 143B-Luciferase control (*n* = 5), or *CNKSR2^{KO}* (*n* = 5) or *SAMD12^{KO}* (*n* = 5) cells. **i** Top panels: Representatives images of livers from the indicated mice at the time of euthanasia are shown. Middle and bottom panels: Representative photomicrographs of H&E-stained liver (left lobe) and lung sections. **j-l** Quantification of macroscopic liver tumours (**j**) and internal metastatic lesions in livers (**k**) or lungs (**l**). Data were obtained from 6 mice (PBS), and 10 mice (control, *CNKSR2^{KO}*, *SAMD12^{KO}*) for each CRISPR clone. One-way ANOVA was performed on data from three independent experiments in (**b**), (**d**), (**f**) and from 6–10 mice in (**j**), (**k**), (**l**). Each treatment group was compared to the Empty vector CRISPR clone 1 control. Error bars correspond to mean values ± SD; ns, not significant (*P* > 0.05). Scale bar: **a**, **c** 200 µm, **e** 500 µm, **i** 5 mm. Source data are provided as a Source Data file.

either with 1 µl of anti-FLAG M2 (Sigma-Aldrich) and 10 µl of protein A/G beads or with 10 µl of GFP-Trap agarose beads (ChromoTek) for 1 h at 4 °C. Beads were then washed at least 3 times with cold lysis buffer and boiled in 2X sample loading buffer (100 mM Tris-HCl pH 6.8, 4% SDS, 0.2% bromophenol blue, 20% glycerol, 200 mM β-mercaptoethanol) for 5 min prior to SDS-PAGE.

For immunoblotting analysis, cells lysates or immunoprecipitated proteins were resolved on SDS–PAGE and transferred onto PVDF membranes (Bio-Rad). Membranes were blocked for at least 30 min in TBST (10 mM Tris-HCl pH 8.0, 150 mM NaCl, 0.2% Tween-20) containing 5% milk (Carnation) or 2% BSA (Sigma-Aldrich) and then incubated with the following primary antibodies for 2 h at RT or overnight at 4 °C: anti-CNK2 from rabbit (1:500; Cedarlane Laboratories; custom-made antibody), anti-α-Tubulin DM1A from mouse (1:10,000; Millipore Sigma; #T9026), anti-CNKSR3 (CNK3) from mouse (1:1000; Abnova; #H00154043), anti-SAMD12 (A-6) from mouse (1:500; Santa Cruz Biotechnology, Inc.; #sc-377123), anti-FLAG M2 from mouse (1:5000; Sigma-Aldrich; #F1804), anti-GFP from rabbit (1:5000; OriGene; #TP401), anti-Phospho-Myosin Light Chain 2 (S19) from rabbit (1:500; Cell Signaling Technology; #3671), anti-Myosin Light Chain 2 from rabbit (1:1000; Cell Signaling Technology; #3672), anti-HGK (MAP4K4) from rabbit (1:1000; Cell Signaling Technology; #3485), anti-TNIK from rabbit (1:1000; GeneTex; #GTX13141), anti-NF2 from rabbit (1:1000; Sigma-Aldrich; #HPA003097), anti-Cytohesin 1 (2E11) (CYTH1) from mouse (1:500; Invitrogen; #MA1-060), anti-Cytohesin-2 (H-7) (CYTH2) from mouse (1:250; Santa Cruz Biotechnology, Inc.; #sc-374640), anti-Cytohesin 3 (CYTH3) from rabbit (1:1000; Abcam; #ab155691); anti-Scribble (SCRIB) from rabbit (1:1000; Cell Signaling Technology; #4475), anti-Cool1/βPix from rabbit (1:1000; Cell Signaling Technology; #4515), anti-β-Arrestin 2 (C16D9) from rabbit (1:1000; Cell Signaling Technology; #3857), anti-RhoA from mouse (1:1000; Cytoskeleton, Inc.; #ARH04), anti-Rac1 from mouse (1:500; Cytoskeleton, Inc.; #ARC03), anti-Cdc42 from mouse (1:500; Cytoskeleton, Inc.; #ACD03), anti-Arf1 from mouse (1:500; Cytoskeleton, Inc.; #ARF01), anti-Arf6 from mouse (1:500; Cytoskeleton, Inc.; #ARF06), anti-Pan Ras from mouse (1:1000; Cytoskeleton, Inc.; #AESA02), anti-Axl (C89E7) from rabbit (1:1000; Cell Signaling Technology; #8661), anti-phospho-Axl (Tyr702) (D12B2) from rabbit (1:1000; Cell Signaling Technology; #5724), anti-GAS6 (D3A3G) from rabbit (1:1000; Cell Signaling Technology; #67202), anti-Akt (pan) (C67E7) from rabbit (1:2000; Cell Signaling Technology; #4691), anti-phospho-Akt (Ser473) (D9E) from rabbit (1:2000; Cell Signaling Technology; #4060), anti-Vimentin (D21H3) from rabbit (1:10,000; Cell Signaling Technology; #5741), anti-SP1 (D4C3) from rabbit (1:1000; Cell Signaling Technology; #9389), anti-MEK1/2 from rabbit (1:2000; Cell Signaling Technology; #9122), anti-Histone H3 (3H1) from rabbit (1:1000; Cell Signaling Technology; #9717), anti-V5 Tag from mouse (1:5000; Invitrogen; #R960-25), anti-His-probe (H15) from rabbit (1:1000; Santa Cruz Biotechnology, Inc.; #sc-803), and anti-Actin (C4) from mouse (1:5000; Millipore Sigma; #MAB1501).

For the secondary antibodies, we used anti-mouse-HRP (1:5000; Jackson ImmunoResearch Laboratories Inc.; #115-035-146) and anti-rabbit-HRP (1:10,000; Jackson ImmunoResearch Laboratories Inc.; #111-035-144). For the detection of biotinylated proteins, we probed the membranes with Streptavidin-HRP (1:20,000; Cytiva; #RPN1231).

## Immunofluorescence procedures and microscopy

U2OS cells were plated in 2-well Culture-Inserts (30,000 cells per insert well; Ibidi; #80209) on 12-mm-diameter coverslips overnight until they reached 100% confluence. Inserts were removed and after 15 h cells were fixed in ice-cold methanol for 5 minutes at RT or 4% paraformaldehyde (PFA) (Electron Microscopy Sciences; #15710) for 15 min at RT. After fixation with PFA, cells were permeabilized with 0.2% Triton X-100 (in PBS) for 5 min. Cells were blocked with 1% BSA (in PBS) for at least 30 min. Primary and secondary antibodies were diluted in 1% BSA (in PBS) and incubated for at least 1 h at RT, and DAPI was added with the secondary antibodies. Cells were washed three times with PBS after both primary and secondary antibody incubations before being mounted on glass microscopy slides in Mowiol medium.

Confocal images of fixed samples were acquired with a Zeiss LSM880 confocal microscope equipped with a Plan-Apochromat 63x/1.4 oil objective controlled with Zen Blue (version 3.0). For each experiment, all images were acquired, extracted, and processed with the same laser settings using the Zen Blue software from Zeiss. All images of Zyxin, pMLC2, and Phalloidin staining were taken at a single ventral focal plane. Inverted and regular LUTs were applied using Zen blue software and exported as uncompressed TIFF files using the Export/Import module. Image cropping was done in ImageJ (version 2.1.0/1.53d).

Mature focal adhesions were detected with the surface module of the Imaris software (Oxford Instruments) version 9.1.2, set on the Zyxin staining channel. Intensity level threshold was applied manually and focal adhesions greater than 1 µm² were considered as mature focal adhesions. For each image, the total number of mature focal adhesions was extracted with the Imaris statistical module and was divided by the total number of cells (DAPI stained nuclei were manually counted on ImageJ). Line scans were performed on the Zen Blue software and plotted in GraphPad Prism 9.0. To obtain a ratio of CNK2 signal at the plasma membrane compared to the cytoplasm, ImageJ was used to measure the CNK2 signal intensity within a 1.32 × 1.32-µm square positioned either in an N-cadherin-positive location (membrane) or an N-cadherin- and DAPI-negative location (cytoplasm). To quantify the percentage of cells displaying FLAG-CYTH1 membrane staining, cells were scored manually for an enrichment of FLAG-staining at the periphery of the cell.

Primary antibodies used for immunofluorescence (IF) were anti-CNK2 produced in rabbit (1:100; Cedarlane Laboratories; custom-made antibody), anti-Phospho-Myosin Light Chain 2 (S19) from rabbit (1:200; Cell Signaling Technology; #3671), anti-Zyxin (2D1) from mouse (1:300; Santa Cruz Biotechnology, Inc.; #sc-293448), anti-N-Cadherin from mouse (1:300; BD Transduction Laboratories; #610921), anti-GFP from rabbit (1:500; OriGene; #TP401) and anti-Flag M2 from mouse (1:1000; Sigma-Aldrich; #F1804).

Secondary antibodies used for IF were goat anti-rabbit A488 (1:250; Invitrogen; #A11008), goat anti-mouse A555 (1:250; Invitrogen;

#A21424) and goat anti-rabbit A555 (1:250; Cell Signaling Technology; #4413).

F-actin was stained with A555 Phalloidin (1:1000; Invitrogen; #A34055) diluted in 1% BSA (in PBS) for 1 h. Biotinylated proteins were stained with Streptavidin-A555 (1:2000; Invitrogen; #S32355) in 1% BSA (in PBS) for 1 h. Nucleic acids were stained with DAPI (Sigma-Aldrich; #MBD0015) in 1% BSA (in PBS) for 1 h.

## Subcellular fractionations

Separation of cytoplasmic, membrane, nuclear soluble, chromatin-bound, and cytoskeletal protein extracts from cultured U2OS cells was performed using the Subcellular Protein Fractionation Kit for Cultured Cells (Thermo Fisher Scientific; #78840), according to the manufacturer's instructions.

## Cell viability assays

Cell viability was assessed using the CellTiter-Glo Luminescent Cell Viability Assay (Promega, Madison MI, USA) every 24 hours for 3 days, according to the manufacturer's instructions. Briefly, cells were seeded into 96-well white flat bottom plates (Falcon; #353296) at a density of 1000 cells/well in 100 µl of culture medium. At the indicated time points, the cell culture medium was removed and 60 µl of CellTiter-Glo reagent was added in each well. The plate was placed on an orbital shaker for 5 min and incubated at room temperature for 5 minutes, and luminescence was measured with a plate reader (BioTek Synergy neo2 multimode reader; Agilent). Data are representative of three independent experiments with two technical replicates for each experiment. Bar graphs represent the ratio of luminescence signal at each time point to the signal at day 0.

## Migration assays

We performed two different types of migration assays: wound healing-like assays using cell culture inserts, or scratch wound healing assays. For wound healing-like assays, 2-well culture inserts (Ibidi; #80209) were placed on a µ-Slide 8-well polymer coverslip (Ibidi; #80826) and U2OS cells ($3 \times 10^4$/well) were plated overnight. Inserts were removed once cells had formed a confluent monolayer, and migration into the artificial wound was imaged with a Zeiss LSM700 confocal microscope equipped with a motorized stage and a stage top incubator (PeCon). Cells were incubated at 37 °C, 5% $CO_2$, and air inside the chamber was humidified. Phase contrast images were taken every 5 min with an N-Achroplan 10x/0.25 PH1 objective controlled by Zen Black (2011) version 1.1.1.0 (Zeiss). Wound area was measured at each time-point using the ImageJ "MRI wound healing tool" plug-in. The difference in wound area between the initial time point and the indicated final time points was used to obtain a percent wound closure. Cells were tracked for at least 8 h using the ImageJ (version 2.1.0/1.53d) "Manual tracking" plugin, and the Ibidi Chemotaxis tool 2.0 was used to calculate cell velocity.

For scratch wound healing assays, cells were seeded overnight ($5 \times 10^4$/well for MDA-MB-231 cells and $3.5 \times 10^4$/well for all other cell lines) in 96-well Incucyte Imagelock (Sartorius; #BA-04857) plates to form a confluent monolayer. Scratch wounds were generated using a wound making tool (Sartorius; #4563) and cells were washed with medium to remove debris. Phase contrast images were taken every hour using the Incucyte S3 live-cell analysis instrument (Sartorius). Data in % wound closure represent the wound confluence metric from the Incucyte analysis software (versions 2020B and 2021C).

## Quantification of cell protrusions

Cell protrusions were quantified using time lapse microscopy images after 10 h of migration in the wound healing assay or using images of fixed cells after 10 h of migration and stained with phalloidin. Protrusion width of cells at the migration front was measured manually using ImageJ. Cells at the migration front were scored manually for the presence of characteristic fan-shaped protrusions. The number of protrusions per cell at the migration front was counted manually.

## Wound healing invasion assays

Incucyte Imagelock (Sartorius; #BA-04857) plates were coated with 0.1 mg/mL Matrigel (Corning; #354234) and incubated at 37 °C overnight. Cells were seeded the following day ($4 \times 10^4$/well for U2OS, HOS and 143B, $4.5 \times 10^4$/well for LN-229, $5 \times 10^4$/well for MDA-MB-231) and incubated for 4–6 h until cells were completely adhered and confluent. The monolayer was wounded as above, and the plate was chilled on ice for 5 min. Culture medium was removed and replaced with 50 µL of ice-cold 2 mg/mL Matrigel diluted in culture medium. To allow the Matrigel to solidify, the plate was incubated at 37 °C for 30 min before the addition of 100 µL of culture medium to each well. Phase contrast images were taken every hour using the Incucyte S3 live-cell analysis instrument. The relative wound density was measured by the Incucyte analysis software (versions 2020B and 2021C).

## Spheroid invasion assays

To form spheroids, $5 \times 10^3$ HOS cells and $2 \times 10^3$ 143B cells were seeded per well into a 96-well ultra-low attachment microplate (Corning; #7007), centrifuged at $125 \times g$ for 10 min at RT, and cultured for 3 days. After spheroid formation, plates were chilled on ice for 10 min before the addition of 100 µL of Matrigel (Corning; #354234) to a final concentration of 4 mg/mL in each well. The plate was incubated at 37 °C for 30 min to allow Matrigel to polymerize and 50 µL of culture medium was then added to each well. Images were taken after addition of Matrigel (Day 0) and 3–4 days later (Day 3 for 143B cells or Day 4 for HOS cells) using a Leica DMIRB inverted microscope and N Plan 5x/0.12 PH0 objective. Spheroid area was measured manually using ImageJ, and the final area (Day 3 or 4) was divided by the initial area (Day 0) to obtain a fold-change for each spheroid.

## Transwell migration and invasion assays

Chemotactic migration (Corning; #354578) and invasion (Corning; #354480) were assessed using transwell inserts with 8 µm pores. After trypsinization, $1 \times 10^5$ cells were resuspended in medium containing 0.5% FBS and seeded into the inserts. The lower chamber contained serum-free medium, medium with 10% FBS, or serum-free medium with 200 ng/mL recombinant human GAS6 (R&D Systems; #885-GSB). Cells were incubated at 37 °C overnight for migration, and for 24 h for invasion (using the Matrigel-coated inserts). Cells were then fixed in cold methanol for 5 min and stained for 10 min with 0.2% crystal violet dissolved in 20% methanol. Images were taken with a Zeiss AXIO imager equipped with a Cannon 5D Mark II camera and LM Scope adaptor. Two fields were imaged for each condition from 3 independent experiments, and the average number of cells per field was calculated.

## GTPase pull-down assays

U2OS cells were seeded in 10 mm petri dishes with 3–4 dishes per experimental sample, and cells were grown to confluence. Scratch wounds (x80) were generated in each dish using an 8-channel aspiration adaptor with p200 micropipette tips[28], and cells were allowed to migrate overnight. Lysis and pull-downs were done using GTPase pull-down activation assay kits (Cytoskeleton; #BK035 (Rac1); #BK036 (RhoA); #BK034 (Cdc42); #BK008 (Ras); #BK032-S (Arf1); #BK033-S (Arf6)) according to the manufacturer's instructions. Briefly, cells were lysed using the provided lysis buffer and equivalent amounts of whole-cell lysate were added to the effector-coupled beads provided with each kit. Lysates with beads were incubated on a rotator at 4 °C for 45 min. Beads were then centrifuged, washed with the provided wash buffer, and resuspended in 2X sample buffer. Western blotting was performed using the provided antibodies and densitometry was conducted manually using ImageJ. Data is presented as a ratio of

GTP-bound (pull-down) to total GTPase (lysate) in each experimental sample. For all pull-downs, ratios were normalized to the control sample.

## Recombinant protein and anti-CNK2 antibody production and purification

The *CNKSR2A* cDNA sequence was codon optimized in the pUC57 plasmid (GenScript, USA) and the first 927 base pairs (amino acids 1-309) were amplified by PCR, cloned into the pPROEX vector (Nco1 and Not1 sites), and sequence verified by Sanger sequencing. This plasmid was then used to transform *E. coli* BL21 (DE3) cells (Invitrogen). Bacterial cultures were grown overnight in LB medium with 100 μg/mL ampicillin and subcultured 1:100 in fresh LB (3 L) for 3 h at 37 °C with 220 rpm agitation until an OD of 0.6 was reached. Expression of TEV-cleavable 6XHis-tagged N-terminal CNK2 proteins was induced by adding 1 mM IPTG to the cell culture. Cells were incubated for 4 h at 37 °C, centrifuged at $4000 \times g$ for 15 min, and stored at −80 °C. To purify N-terminal CNK2 proteins, cell pellets were resuspended in ice-cold lysis buffer (100 mM Tris-HCl pH 7, 500 mM NaCl, 25 mM imidazole, 10% glycerol, 1 mM PMSF, 5 μg/mL DNase I) and sonicated for 10 min at 4 °C. After centrifugation at $45,000 \times g$ for 1.5 h, the supernatant was filtered and loaded onto a 5 ml Ni-NTA agarose column (Qiagen) pre-equilibrated with lysis buffer. The column was subsequently washed with 50 mL of wash buffer (100 mM Tris-HCl pH 7, 500 mM NaCl, 50 mM imidazole, 10% glycerol). Proteins were eluted with 15 mL of wash buffer containing 500 mM imidazole. Proteins fractions (1 mL each) were then analyzed by SDS-PAGE followed by TEV protease treatment and overnight dialysis in 100 mM Tris-HCl pH 7, 500 mM NaCl, 10% glycerol. Proteins were again loaded onto the Ni-NTA agarose column to remove cleaved His-tag peptides and non-cleaved proteins. Cleaved proteins in the flow-through were concentrated using Amicon filters (Millipore Sigma, Cut-off 3 kDa), frozen in liquid nitrogen, and stored at −80 °C. TEV cleavage efficiency was verified by SDS-PAGE and protein authenticity was confirmed by mass spectrometry. Purified N-terminal CNK2 was then sent to Cedarlane Laboratories (Canada) and used to immunize rabbits.

Anti-CNK2 antibodies from the rabbit polyclonal serum were purified by affinity chromatography. Briefly, 7.5 mg of purified N-terminal CNK2 (amino acids 1-309) was dialyzed in coupling buffer (100 mM NaHCO$_3$, 500 mM NaCl pH8.3) and incubated with 6 mL of the CNBr Activated Sepharose 4B beads overnight at 4 °C. The beads were then blocked with 100 mM Tris pH 8.0 and washed three times in tandem with the two buffers: (1) 500 mM NaCl, 100 mM sodium acetate pH 3.5, and (2) 500 mM NaCl, 100 mM Tris pH 8.5. The beads were then resuspended in PBS pH 7.5 and mixed with the polyclonal antiserum. After overnight incubation at 4 °C with low agitation, beads were washed with 60 mL of PBS. Bound antibodies were eluted using 100 mM glycine pH 2.5 and neutralized with 100 mM Tris pH 10 (ratio 10:1). Collected antibodies were then dialyzed overnight in PBS, concentrated to 1 mg/mL, aliquoted, flash frozen on dry ice and stored at −80 °C.

Recombinant CNK2 PH domain proteins (His-CNK2$^{553-699}$ WT and His-CNK2$^{553-699}$_W591A), used in PIP Strips assays (see next section), were expressed as described above (except lysis buffer contained 0.5% Triton X100) in *E. coli* BL21 (DE3) cells, purified with nickel beads and eluted with imidazole.

## PIP strips

Purified CNK2 protein fragments encompassing the PH domain, His-CNK2$^{553-699}$ WT and His-CNK2$^{553-699}$_W591A, were used to probe PIP Strips (Echelon Biosciences; #P6001) for lipid binding, according to the manufacturer's instructions. Each PIP Strip consists in a 2 cm × 6 cm hydrophobic membrane that has been spotted with 15 different lipids at 100 pmol per spot. PIP Strips were blocked in PBS containing 3% BSA and 0.1% Tween 20 (PBST) for one hour at RT. The membranes

were then incubated in blocking solution supplemented with 2 μg/mL recombinant His-CNK2 PH domain (WT or W591A) for one hour at RT with gentle agitation. After removing the protein solution and washing three times with PBST, the membrane was incubated with anti-His antibody for 1 h followed by HRP-conjugated secondary antibody. His-CNK2 PH domain binding was detected by enhanced chemiluminescence (ECL).

## Treatment with inhibitors and AXL stimulation with GAS6

Receptor tyrosine kinases (RTKs) were inhibited in U2OS cells overnight with 1 μM of each of the following inhibitors: Gefitinib (#HY-50895), Lapatinib (#HY-50898), Bemcentinib (#HY-15150), BMS-536924 (#HY-10262) purchased from MedChemExpress; Erdafitinib (#HY-18708) Lenvatinib (#HY-10981), Cabozantinib (#HY-13016), Galunisertib (#HY-13226), Linsitinib (#HY-10191) from Cedarlane Laboratories; and Crizotinib (#PF-02341066) from Selleckchem.com. PI3K was inhibited by treating cells with LY294002 (Millipore Sigma; #440202) at 10 μM for at least 1 h. Gas6-mediated stimulation of AXL in U2OS cells was performed by serum-starving cells for 24 hours prior to the addition of 200 ng/mL recombinant Human Gas6 (R&D Systems; #885-GSB) to the culture medium for 30 min. For AXL receptor blockade, U2OS cells were incubated with 20 μg/mL anti-AXL antibody (R&D Systems; #AF154) or 50 μg/mL anti-GAS6 antibody (R&D Systems; #AB885). Normal goat polyclonal IgG (R&D Systems; #AB-108-C) was used as control (50 μg/mL). Rho-associated kinases (ROCK) were inhibited with Y-27632 (Cell Signaling Technology; #13624) at 1 or 5 μM.

## RNA extraction and RT-qPCR

Total RNA was extracted from cells and purified using either the RNeasy Mini Kit (Qiagen; #74104) following the manufacturer's instructions or using standard TRIzol procedures. For TRIzol extraction, cells were washed with cold PBS and lysed directly in the culture dishes by adding TRIzol (Life Technologies; #15596026). 0.2 mL of chloroform (Fisher Scientific; #BP1145-1) per 1 mL of TRIzol was then added and tubes were vortexed for 15 s and incubated at RT for 2 min. The cell lysate was then spun at $22,000 \times g$ for 15 min at 4 °C. The aqueous phase was recovered and transferred to a new tube and the RNA was precipitated by adding 0.5 mL of 2-propanol (Fisher Scientific; #A416P-4) per 1 mL of TRIzol and incubated at RT for 10 min. Samples were then spun at $11,000 \times g$ for 10 min at 4 °C and the RNA pellet was washed with 1 mL of 75% EtOH. Finally, the RNA pellet was spun at $6300 \times g$ for 5 min at 4 °C and air-dried for 5–10 min, resuspended in RNAse free H$_2$O, and stored at −80 °C. RNA integrity was validated using a Bioanalyzer 2100 (Agilent). Total RNA was treated with DNase and reverse transcribed using the Maxima First Strand cDNA synthesis kit with dsDNase (Thermo Fisher Scientific). Gene expression was determined using assays designed with the Universal Probe Library from Roche (www.universalprobelibrary.com). Oligonucleotide sequences are listed in Supplementary Data 4 in the "RT-qPCR oligos" tab. For each qPCR assay, a standard curve was performed to ensure that the efficiency of the assay was between 90% and 110%. A QuantStudio qPCR system (Thermo Fisher Scientific) was used to determine amplification levels. Relative expression comparison (RQ = 2^-ΔΔCT) was calculated using Expression Suite version 1.3 (Thermo Fisher Scientific), using both GAPDH and ACTB as endogenous controls.

## Generation of CRISPR/Cas9 knockout cells

Guide RNAs targeting the genomic sequences of human *CNKSR2* and *SAMD12* (Supplementary Data 4, "Guide RNA oligos" tab) were designed using the CRISPick sgRNA Design tool from the Broad Institute and were verified for specificity using the NCBI BLAST tool. Guide RNA oligonucleotides (synthesized from Sigma-Aldrich) were cloned in the pLentiCRISPRv2 vector (Addgene; #52961) at the BsmBI cloning site. For CRISPR knockout (KO) cell generation, dilution cloning of

143B cells was performed to homogenize the genetic background. The original pLentiCRISPRv2 or vectors containing the guide RNA targeting *CNKSR2* or *SAMD12* were then used for lentiviral production and infection of the clonal 143B cell line, as described in the lentiviral infections section. 24 h after infections, 143B cells were selected with 2.5 µg/mL puromycin for 24 h. Dilution cloning of surviving cells was then performed to amplify individual CRISPR clones. Finally, *CNKSR2* and *SAMD12* CRISPR KO clones were selected following western blotting and RTq-PCR validation of the KO efficiency.

## Animal experiments

All mice were bred under standard conditions at the Institute for Research in Immunology and Cancer. Immunodeficient NOD.Cg-*Prkdc*$^{scid}$ *Il2rg*$^{tm1Wjl}$/SzJ mice (NSG) were originally obtained from Jackson Laboratory (RRID:IMSR_JAX:005557). Mice were housed under specific pathogen-free conditions in ventilated racks within filter-topped isolator cages, with a 12/12 h light/dark cycle, at constant temperature (20 °C +/− 2 °C) and humidity (50% +/− 10%), and with access to food and water *ad libitum*. Ethical endpoints accepted by the ethics committee were tumour burden as determined by a positive PAAM test (Pulmonary assessment of advanced metastasis), or weight loss of 20%. A pilot study had ensured that these endpoints were not reached in the 3-week interval in the experimental conditions described and no animals reached these endpoints during the study.

## Tumour xenograft experiments and bioluminescence imaging

For experimental metastasis assay, $1 \times 10^5$ 143B osteosarcoma cells resuspended in 200 µL PBS were injected in the lateral tail vein of 14–17-week-old male NSG mice. Tumour development in mice injected with luciferase-expressing 143B cells was quantified by bioluminescence imaging once per week for 3 weeks. In brief, mice were injected intraperitoneally with a 30 mg/mL solution of D-luciferin (MediLumine) diluted in PBS at a dose of 150 mg/kg or approximately 4.5 mg/mouse 20 min before imaging. Mice were then anesthetized with 2.5% isoflurane and maintained under anesthesia until the end of image acquisition. Mice were imaged with LabeoTech OiS300 In Vivo Imaging System (Labeo Technologies Inc.). Signal normalization and analysis was done automatically for all time points using ImageJ (version 2.1.0/1.53d) macros and expressed in radiance (photons•s$^{-1}$•sr$^{-1}$•cm$^{-2}$) integrated density (Area•mean intensity).

## Histology

Mice were euthanized 3 weeks after intravenous injection of 143B cells in the tail vein. Livers and lungs were collected and fixed in 10% formalin for 24 h at RT. All liver surface lesions were counted manually. Left lateral lobes of livers and whole lungs were embedded in paraffin and sliced into 5-µm-thin sections spaced 400 µm apart. Tissue sections were mounted on glass slides and stained with hematoxylin and eosin (H&E) by conventional protocols. Internal lesions in the left liver lobe and the lungs were counted manually using the NDP.view2Plus software (version 2.9.29). Two different slices were used for the left liver lobe, and one slice for the lungs.

## Statistics and reproducibility

All statistical analyses were conducted with GraphPad Prism 9.0 (Graph Pad Software Inc.). We used unpaired two-tailed Student's *t*-test, or one-way or two-way analysis of variance (ANOVA), as indicated in the corresponding figure legends. All graphs show the mean values ± SD. In all figures, $P > 0.05$ was considered not significant (ns). We used at least three independent experiments or six biologically independent samples (mice) for statistical analysis. PIP Strip assays, subcellular fractionations, and immunoblots or confocal images without associated quantifications were conducted independently at least twice. All other experiments in the manuscript were performed at least three times independently, with similar results. No statistical method was used to pre-determine sample size. No data were excluded from the analyses. The investigators were not blinded during data analyses.

## Reporting summary

Further information on research design is available in the Nature Portfolio Reporting Summary linked to this article.

## Data availability

Publicly available Gene Ontology terms used to classify BioID hits (Supplementary Fig. 1e; Supplementary Fig. 3d) are available on STRING (https://string-db.org/). Comparative expression of *CNKSR1, CNKSR2, CNKSR3 and IPCEF1* genes in healthy human tissues were acquired from the Genotype-Tissue Expression (GTEx) database on 02/01/22 using the Multi-Gene Query expression option (https://gtexportal.org/home/). The mass spectrometry proteomics data generated in this study (Supplementary Data 1 and 3) are available via the ProteomeXchange Consortium through the PRIDE partner repository under the accession code PXD038687. The remaining data are available within the Article, Supplementary Information or Source Data files. Source data are provided with this paper.

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

## Acknowledgements

We thank A.C. Gingras, J. F. Côté, P. Roux, K. L. B. Borden, J. Parvin and S. Blacklow for their generous gift of DNA constructs and cell lines, and C. Baril and H. Lavoie for critical reading of the manuscript. We are also grateful for J. F. Côté and P. Roux for advice on BioID experiments, E. Bonneil from the Proteomics facility, C. Charbonneau from the Bio-Imaging facility, and all members from Histology, Genomics, and In Vivo Biology facilities at the Institute for Research in Immunology and Cancer at Université de Montréal. Proteomics analyses were performed by the Centre for Advanced Proteomics Analyses, a Node of the Canadian Genomic Innovation Network that is supported by the Canadian Government through Genome Canada. J. Gagnon held a doctoral scholarship from Fonds de Recherche du Québec – Santé (FRQS). This work was supported by a Foundation grant (FDN-388023) from the Canadian Institutes for Health Research (CIHR) to M.T., by a Project grant from the CIHR to G.E. (PJT-175093), and by a grant from the Cancer Research Society to S.M. (25388). M.T. holds a Tier 1 Canada Research Chair in Intracellular Signalling.

## Author contributions

G.S, D.K., J.G., C.P., M.K.S.E.L., and M.T. designed the experiments. G.S, D.K., J.G., and M.T. wrote the manuscript. G.S, D.K., J.G., C.P., D.L., E.D., M.S., D.G., M.L., G.A., and M.K.S.E.L. conducted the experiments. S.M. provided guidance with the animal studies. G.E. provided guidance with cell biology studies.

## Competing interests

The authors declare no competing interests.
