## [Peer Review File · Nature Communications]

Reviewer comments, first round

Reviewer #1 (Remarks to the Author):

In this study, the authors characterize a conserved signaling pathway that is important for motility in a number of distinct cancer cell types *in vitro*, and at least one *in vivo*. In this pathway, activation of the receptor tyrosine kinase AXL leads to the PI3K-dependent recruitment of a protein complex containing the scaffolding protein CNK2, the adaptor protein SAMD12 and two activators of the ARF6 GTPase, cytohesin-1 and cytohesin-3. Previous work by others has shown that ARF6 activation promotes the downstream activation of the Rho family GTPase Rac1, and that this is essential for cell motility. Here the authors provide convincing evidence that knockdown of CNK2, SAMD12 or the cytohesins attenuates ARF6 activation, downstream Rac1 activation and consequently cell migration in both 2D and 3D assays. Importantly, this regulatory mechanism also appears to function *in vivo*; CRISPR-mediated knockout of either CNK2 or SAMD12 results in a dramatic reduction in metastases in a mouse xenograft model of osteosarcoma without affecting growth or survival of the primary tumor. It has been known for some time that cytohesins promote motility by stimulating an ARF6/Rac1 activation cascade, but the requirement for CNK2 and SAMD12 in this context is novel and is an important contribution to the field.

In general the study is well-designed, with all necessary controls and appropriate statistical analysis throughout. The results are clear and support the conclusions drawn by the authors. There are, however, several issues that remain to be adequately addressed:

1. Fig. 1 d-h show that knockdown of either CNK2 or CNK3 impairs migration in some, but not all cell lines tested. This doesn't appear to be due to relative expression levels (Fig. 1C). How do the authors interpret this observation?
2. Knockdown of cytohesin-1 or -3 individually does not inhibit migration as potently as a double knockdown (Fig. 3C). This suggests at least partial redundancy, which should be addressed more directly in the text. Interestingly, cytohesin-2 does not appear to play a role in this pathway, although it has been shown to promote migration in other cell types. The cytohesins exhibit an interesting alternative splicing event that alters their affinity for phosphoinositides, the difference being the presence or absence of a single glycine residue in the PH domain (Cronin et al, EMBOJ 2004). Do the authors know which splice variant(s) the 3 expressed isoforms represent? It is plausible that cytohesin-1 and -3 are PIP3-selective variants, while cytohesin-2 is the less-selective variant that would be less influenced by PI3K activity. This could be determined by either RNAseq or by sequencing PCR products.
3. Fig. 4i, j show that knockdown of either cytohesin-1 or cytohesin-3 substantially inhibits Rac1 activation, but that only the double knockdown results in increased RhoA activity. How do the authors explain this?
4. In their discussion, the authors state that "identification of ARF6-regulated enzymes directly inhibiting RhoA is also an appealing prospect". There are several ARF6 GAPs that also contain RHO-GAP domains, ARAP1, ARAP2 and ARAP3 (also known as centaurin-1-3). As GAPs can sometimes serve as effectors of their target GTPases, it is plausible that one or more ARAPs serve this function. This should be mentioned in the discussion.

Reviewer #2 (Remarks to the Author):

This is an impressive work that explores the functional activity and role of CNK family protein particularly focusing on CNK2. Through BioID-based proteomics, CNK2-specific interactors are identified and used to guide the search for cellular processes in which CNK2 is involved. This led the authors to investigate the role of CNK2, through loss and gain of function approaches, in cell migration and invasion. They further exploited the set of identified interactors to define possible pathways underlying CNK2 function in migration/invasion. These experiments revealed that CNK2 acts at the plasma membrane downstream of AXL to promote via cytohesins the activation of ARF6 and the ensuing elevation of RAC1GTP, in turn, required for promoting enhanced cell migration.

They further investigate the role in the process of SAMD12, which is also involved in cell migration and tumor dissemination, albeit the mechanisms of its action remain unclear.

This is an almost "classical type of cell biological work" that exploits standard biochemistry, and cell biology to build the possible components of new CNK2-ARF6 centric pathway with a potential impact on tumor cell dissemination. In this respect, the work is conducted logically and rigorously with no major flaws and several complementary approaches. The authors should certainly be commended for having been able to decode an entirely novel signaling axis.

There are, however, several questions that the work arises, and addressing some of them may increase the relevance of the entire manuscript.

Firstly, the work demonstrates the role of AXL in promoting CNK2-mediated ARF6 activation and migration/invasion. However, it is unclear during migration and invasion whether GAS6/AXL are activated at all and do mediate migratory processes.

Secondly, it remains entirely unclear how CNK2 is regulated by AXL and, in turn, regulates Cytohesin. For example, some evidence is produced that CNK2 PM localization and redistribution is key for its activity, yet it remains unclear whether and how it acts on Cytohesins at the plasma membrane. Does it activate their GEF activity? Admittedly, exploring this aspect might be part of the next installments but some discussion or even experiments on this point would be relevant.

Thirdly, the migratory/invasion assays are well conducted, but these days are suboptimal. Most of the assays done are in wound healing which measures collective locomotory behavior. Under these conditions, the loss of CNK2 robustly inhibits wound closure. A superficial look at the cell behavior at the wound front seems to indicate that the loss of CNK2 drastically impairs the ability of cells of extending cell protrusion (some quantification of the activity of cells at the front might provide some support to this possibility), which would be consistent with impairments of RAC1 activity and the observed increased adhesion. One might predict that individual cell locomotion and especially locomotion toward GAS6 is severely affected by CNK2 loss of function. This can and should be directly tested and may provide the basis to link GAS6-AXL with CNK2-ARF6 pathway. Similarly, there might be an impact on individual cell invasion which cannot be inferred from the spheroid assays invasion as presented. Stated differently, the authors appear in the optimal condition to explore the impact of the GAS6/AXL/CNK-ARF6 axis on individual cell locomotion and invasion with little extra effort.

Finally, the experimental metastasis assays are convincing. Yet it remains entirely unclear whether CNK2 is specifically deregulated in solid tumors, or is simply required for basic migratory/invasive activity.

Reviewer #3 (Remarks to the Author):

The manuscript entitled "The scaffold protein CNK2 promotes cancer cell motility by mediating ARF6 activation downstream of AXL signalling" characterizes the scaffold proteins of the CNK family from human cells, and identifies a role of CNK2 in promoting cancer cell migration through coupling of AXL to the activation of ARF6 GTPase, including a novel adaptor protein called SAMD12. This activation of ARF6-GTP controls motile forces through RAC1 and RHOA GTPases. Mouse xenograft models with CNK2 and SAMD12 KO reduces metastasis, confirming the importance of this pathway in motility of cancer cells and potentially open new avenues for targeting metastasis. The paper thus uncover a previously uncharacterized mechanism of regulation cell motility through a family of scaffold proteins that were poorly characterized up to now.

The authors initially defined the proximal interactome of four human CNKs, in both inducible 293 and U2OS cells using BioID. They identified several interactions including known interactors. Clustering and GO annotations on biological processes revealed enrichments in cell junction organization, cytoskeleton organization, cell polarity and adhesion, although these enrichments are not shown (lanes 88-91), neither is the network of candidates in U2OS cells (lanes 138-139). This should be provided as a table ranked with the enrichment score to have a better view of the functions of the proteins identified. A network with known interactions should also be provided to visualized known interactions in these proteins.

The authors then focus on CNK2 based on the high levels of expression of this gene in a number of

cancer cells, including two osteosarcoma cell lines despite lack of expression in normal osteoblast cells. They then characterized how CNK2 regulates RAC1 and RHOA GTPases during migration in a number of assays, and demonstrate that CNK2 operates at the plasma membrane through a phosphoinositide-binding PH domain.

Overall, the authors have demonstrated a CNK2-dependant signalling axis triggered by AXL that is involved in cancer cell migration, invasion and metastasis. This is a significant discovery and characterization of an important pathway.

One interesting question that remains is whether any of these proteins are direct targets for phosphorylation by the RTK activated (or other kinases), and what other proteins found in proximity are regulated in these conditions. The authors potentially already have this data, since the proteomic analyses included the S/T/Y phosphorylation sites. Since this data has already been extracted, it should at least be mentioned if any relevant modifications were identified. The phosphopeptides information is not provided.

Other minor suggestions:

Some details are missing from the methods section. In particular, which uniprot database was used? Only the number of entries is provided. The raw files and data should be submitted in a repository for access.

Some of the figures are very small and difficult to see without extensive zooming of the PDF. For example, the contrast is almost impossible to see in Figure 1d, but much easier to see in Figure 1k.

Lane 43: there may be a reference missing with reference 6?

Response to Reviewers

We would like to thank the reviewers for taking the time to read our manuscript and for their positive feedback and constructive criticism. We have considered their comments and made appropriate changes to produce a revised version of our original manuscript. Our responses to each comment (reported *verbatim*) are outlined below.

Reviewer #1 - ARF6 signaling

In this study, the authors characterize a conserved signaling pathway that is important for motility in a number of distinct cancer cell types *in vitro*, and at least one *in vivo*. In this pathway, activation of the receptor tyrosine kinase AXL leads to the PI3K-dependent recruitment of a protein complex containing the scaffolding protein CNK2, the adaptor protein SAMD12 and two activators of the ARF6 GTPase, cytohesin-1 and cytohesin-3. Previous work by others has shown that ARF6 activation promotes the downstream activation of the Rho family GTPase Rac1, and that this is essential for cell motility. Here the authors provide convincing evidence that knockdown of CNK2, SAMD12 or the cytohesins attenuates ARF6 activation, downstream Rac1 activation and consequently cell migration in both 2D and 3D assays. Importantly, this regulatory mechanism also appears to function *in vivo*; CRISPR-mediated knockout of either CNK2 or SAMD12 results in a dramatic reduction in metastases in a mouse xenograft model of osteosarcoma without affecting growth or survival of the primary tumor. It has been known for some time that cytohesins promote motility by stimulating an ARF6/Rac1 activation cascade, but the requirement for CNK2 and SAMD12 in this context is novel and is an important contribution to the field.

In general the study is well-designed, with all necessary controls and appropriate statistical analysis throughout. The results are clear and support the conclusions drawn by the authors. There are, however, several issues that remain to be adequately addressed:

We thank reviewer #1 for mentioning our important contribution to the field and for recognizing the quality of our work.

1. Fig. 1 d-h show that knockdown of either CNK2 or CNK3 impairs migration in some, but not all cell lines tested. This doesn't appear to be due to relative expression levels (Fig. 1C). How do the authors interpret this observation?

This is an interesting question regarding the cell line-dependent effect of CNK3 knockdown on cell migration but not invasion. We did not address this point in our manuscript because, as CNK2 knockdown had the most widespread effect on motility across several cell lines, the large amount of data already present in the current manuscript, and the uncertainty of discovering the underlying cause, we decided to focus only on the

molecular characterization of CNK2 (not CNK3). Evidently, various hypotheses could account for the apparent cell line dependency on CNK3 for cell migration.

(1) It is possible that for some cell lines, the migration assay is not sensitive enough for the detection of a CNK3-dependent effect. Since the wound is closed much quicker in the migration assay compared to the invasion assay, it is possible that more time is required to observe an effect of CNK3 depletion during migration for certain cell lines. (2) Since one of the limitations of these experiments is the incomplete knockdown achieved using shRNA, this would also be consistent with a remaining fraction of CNK3 being sufficient to allow motility early on during these assays, but not over the course of the longer invasion assay. (3) The requirement for CNK3 during migration could vary between cell lines depending on the substrate type. As all our migration experiments were conducted on tissue culture-treated plastic, it is possible that cells migrating on different substrates, such as a Matrigel coating, would be more sensitive to CNK3 depletion. (4) Finally, CNK3 could have an important pro-invasion function that is not captured by the migration assay, in which case some cell lines might be dependent on this pro-invasion function without requiring CNK3 for migration.

2. Knockdown of cytohesin-1 or -3 individually does not inhibit migration as potently as a double knockdown (Fig. 3C). This suggests at least partial redundancy, which should be addressed more directly in the text.

This is a good point raised by the reviewer. Indeed, the inhibition of migration caused by double knockdown of CYTH1 and CYTH3 was significantly greater than that resulting from either individual depletion. We agree that our findings suggest a partial redundancy between CYTH1 and CYTH3 function during cell migration. We have addressed this point in the Discussion section (**lines 435-436**).

3. Interestingly, cytohesin-2 does not appear to play a role in this pathway, although it has been shown to promote migration in other cell types. The cytohesins exhibit an interesting alternative splicing event that alters their affinity for phosphoinositides, the difference being the presence or absence of a single glycine residue in the PH domain (Cronin et al, EMBOJ 2004). Do the authors know which splice variant(s) the 3 expressed isoforms represent? It is plausible that cytohesin-1 and -3 are PIP3-selective variants, while cytohesin-2 is the less-selective variant that would be less influenced by PI3K activity. This could be determined by either RNAseq or by sequencing PCR products.

Indeed, it is an intriguing finding that CYTH1 and CYTH3, but not CYTH2, are required for cell migration in U2OS cells. As mentioned, this could be explained by the relative abundance of respective di-glycine (more PIP3 responsive) and tri-glycine (more PIP2 responsive) splice variants for each *CYTH* gene. To this end, as suggested by the reviewer, we have sequenced PCR products generated from U2OS-derived cDNAs and compared

chromatograms to a plasmid reference sequence to quantify the percentage of transcripts containing the third Gly residue for each gene. We determined % di-glycine values of 27.4% for CYTH1, 39.8% for CYTH2, and 91.8% for CYTH3 (see graph on the right). These results therefore suggest that the majority of CYTH1 and CYTH2 are the PIP2 selective variants, whereas CYTH3 is mostly expressed as the PIP3 selective variant. However, more work would be necessary to characterize if CYTH1 and CYTH3 variants are differentially required for CNK2-dependent signalling during migration. As our study characterizes a novel function for CNK2 in cancer cells, we feel it to be slightly out of scope to further investigate Cytohesin splice variants in this manuscript and therefore respectfully request to leave this topic for another study.

4. Fig. 4i, j show that knockdown of either cytohesin-1 or cytohesin-3 substantially inhibits Rac1 activation, but that only the double knockdown results in increased RhoA activity. How do the authors explain this?

Indeed, the increase in RHOA activity caused by either knockdown of CYTH1 or CYTH3 (Fig. 4j) appears more modest on the western blot than the decrease of RAC1 activity (Fig. 4i). Nevertheless, when quantified, the increase in RHOA activity is statistically significant, and thus agrees with our proposed model. Since the RAC1 and RHOA activation assays use different effector-coupled beads to purify each respective GTPase, it is possible that the more modest impact on RHOA activity (as it appears on the western blot) is due to the experimental conditions imposed by the assay. In addition, the observed difference is also consistent with the partial functional redundancy observed between CYTH1 and CYTH3, with the combined knockdown of CYTH1 and CYTH3 having a greater effect on RHOA activation than either alone.

5. In their discussion, the authors state that “identification of ARF6-regulated enzymes directly inhibiting RhoA is also an appealing prospect”. There are several ARF6 GAPs that also contain RHO-GAP domains, ARAP1, ARAP2 and ARAP3 (also known as centaurin-1-3). As GAPs can sometimes serve as effectors of their target GTPases, it is plausible that one or more ARAPs serve this function. This should be mentioned in the discussion.

We thank the reviewer for this comment. We agree that ARAPs could serve as ARF6-regulated factors that mediate inhibition of RHOA. Accordingly, we have addressed this point in the Discussion (lines 483-486).

Reviewer #2 - Cell migration, Rho/Rac

This is an impressive work that explores the functional activity and role of CNK family protein particularly focusing on CNK2. Through BioID-based proteomics, CNK2-specific interactors are identified and used to guide the search for cellular processes in which CNK2 is involved. This led the authors to investigate the role of CNK2, through loss and gain of function approaches, in cell migration and invasion. They further exploited the set of identified interactors to define possible pathways underlying CNK2 function in migration/invasion. These experiments revealed that CNK2 acts at the plasma membrane downstream of AXL to promote via cytohesins the activation of ARF6 and the ensuing elevation of RAC1GTP, in turn, required for promoting enhanced cell migration. They further investigate the role in the process of SAMD12, which is also involved in cell migration and tumor dissemination, albeit the mechanisms of its action remain unclear.

This is an almost “classical type of cell biological work” that exploits standard biochemistry, and cell biology to build the possible components of new CNK2-ARF6 centric pathway with a potential impact on tumor cell dissemination. In this respect, the work is conducted logically and rigorously with no major flaws and several complementary approaches. The authors should certainly be commended for having been able to decode an entirely novel signaling axis.

We thank Reviewer #2 for their kind words and acknowledging our identification of a novel signaling axis.

There are, however, several questions that the work arises, and addressing some of them may increase the relevance of the entire manuscript.

Firstly, the work demonstrates the role of AXL in promoting CNK2-mediated ARF6 activation and migration/invasion. However, it is unclear during migration and invasion whether GAS6/AXL are activated at all and do mediate migratory processes.

The role of GAS6 and AXL in promoting cell motility has been previously established, and notably, AXL was found to promote motility in osteosarcoma and glioblastoma cells^{1,2}. Our study supports these findings since we reported in the original manuscript that AXL depletion by shRNA impairs cell migration in U2OS, HOS, and LN-229 cells in the wound healing assay (**Original Extended Data Fig. 6b, c**). However, we agree that our experiments did not thoroughly address whether AXL is activated during migration and invasion, and whether GAS6 also participates in these processes. Since AXL can be activated through multiple mechanisms including ligand-dependent and -independent homo- or hetero-dimerization, we wanted to investigate whether GAS6 indeed contributes to AXL activation during cell motility in U2OS cells.

First, to test the requirement for GAS6 and AXL in U2OS cell migration and invasion, we used blocking antibodies against GAS6 or AXL to block receptor-ligand interactions and conducted wound healing migration and invasion assays. We found that the addition of

either blocking antibody led to a decrease in AXL activation assessed by measuring pTyr702 on AXL and inhibited both cell migration and invasion of U2OS cells (**Revised Supplementary Fig. 8d-h**). Additionally, we tested the effect of GAS6 knockdown on U2OS and LN-229 cell migration and invasion using two different shRNAs. These experiments revealed that, like depletion of AXL, GAS6 depletion strongly inhibits migration and invasion in multiple cell lines (**Revised Supplementary Fig. 8i-m**), indicating that secreted GAS6 stimulates motility in an autocrine and/or paracrine manner.

Finally, to further test whether GAS6 induces migration, we stimulated serum starved U2OS cells with 200 ng/ml of recombinant GAS6 added to the cell culture medium and conducted a wound healing migration assay. We found that this treatment induces cell migration, and strikingly, GAS6-induced migration was not only suppressed by knockdown of AXL, but also by knockdown of CNK2, demonstrating that AXL and CNK2 are required for GAS6-induced migration in U2OS cells (**Revised Supplementary Fig. 9a-c**). Together, these results provide strong evidence that AXL is indeed activated by GAS6 to stimulate cell motility in a CNK2-dependent manner.

To account for these new findings, we have modified the text (**lines 360-374**).

The Methods section has also been modified accordingly to account for the new reagents and experimental procedures.

Ref 1 : Han, J. et al. Gas6/Axl mediates tumor cell apoptosis, migration and invasion and predicts the clinical outcome of osteosarcoma patients. *Biochemical and Biophysical Research Communications* 2013, doi:10.1016/j.bbrc.2013.05.019.

Ref 2: Zdzalik-Bielecka, D. et al. The GAS6-AXL signaling pathway triggers actin remodeling that drives membrane ruffling, macropinocytosis, and cancer-cell invasion. *Proc Natl Acad Sci U S A* 118, doi:10.1073/pnas.2024596118 (2021).

Secondly, it remains entirely unclear how CNK2 is regulated by AXL and, in turn, regulates Cytohesin. For example, some evidence is produced that CNK2 PM localization and redistribution is key for its activity, yet it remains unclear whether and how it acts on Cytohesins at the plasma membrane. Does it activate their GEF activity? Admittedly, exploring this aspect might be part of the next installments but some discussion or even experiments on this point would be relevant.

Indeed, our study suggests that CNK2 enhances the presence of CYTH1 at the plasma membrane, and thus likely recruits it to specific regions where ARF6 may become activated. However, we did not elaborate on the mechanism by which their interaction at the plasma membrane might promote ARF6 activation. It is possible that binding with CNK2 enhances the GEF activity of CYTH1 and/or CYTH3 toward ARF6 through an allosteric mechanism, or that another binding partner of CNK2 regulates Cytohesin GEF activity. Conversely, Cytohesins could be activated independently of CNK2, and the

function of CNK2 may be limited to bringing Cytohesins into proximity with ARF6 at the plasma membrane. We have added to our Discussion on this point in the manuscript (**lines 436-441**).

Thirdly, the migratory/invasion assays are well conducted, but these days are suboptimal. Most of the assays done are in wound healing which measures collective locomotory behavior. Under these conditions, the loss of CNK2 robustly inhibits wound closure. A superficial look at the cell behavior at the wound front seems to indicate that the loss of CNK2 drastically impairs the ability of cells of extending cell protrusion (some quantification of the activity of cells at the front might provide some support to this possibility), which would be consistent with impairments of RAC1 activity and the observed increased adhesion.

We thank the reviewer for this comment regarding the visible defect in protrusion formation. We agree that our time lapse microscopy experiments (**Fig. 1k and Supplementary Movie 1**), as well as the finding that CNK2-depleted cells display less active RAC1 (Fig. 2a), suggest a defect in the ability of these cells to generate lamellipodia at the migration front. To address this point, we quantified protrusion width and scored the shape of the protrusions in control cells and cells depleted of CNK2 by shRNA (**Revised Supplementary Fig. 4b-e**). This analysis revealed that knockdown of CNK2 results in significantly fewer cells extending fan-like lamellipodia at the migration front. Instead, cells lacking CNK2 often display protrusions that are thinner, and they also possess a greater number of protrusions, indicating a defect in forming a lamellipodium. These defects were rescued by exogenous expression of CNK2A but not CNK2B (which lacks the cytohesin-binding domain). In agreement with these findings, we also stained migrating cells with phalloidin and observed that CNK2 knockdown resulted in fewer cells at the migration front displaying lamellipodia, in similar proportions to that observed by time-lapse microscopy (**Revised Supplementary Fig. 4f, g**). To account for these new results, we modified the text accordingly (**lines 157-168**).

The Methods section has also been modified accordingly to account for the new experimental procedures.

One might predict that individual cell locomotion and especially locomotion toward GAS6 is severely affected by CNK2 loss of function. This can and should be directly tested and may provide the basis to link GAS6-AXL with CNK2-ARF6 pathway. Similarly, there might be an impact on individual cell invasion which cannot be inferred from the spheroid assays invasion as presented. Stated differently, the authors appear in the optimal condition to explore the impact of the GAS6/AXL/CNK-ARF6 axis on individual cell locomotion and invasion with little extra effort.

This is an interesting point regarding the role of CNK2 on single cell migration and chemotaxis. Indeed, the wound healing migration and invasion assays, and the spheroid invasion assays, do not assess the necessity for CNK2 in single cell motility. We also agree that CNK2 may be required for chemotaxis toward GAS6 since our data indicate that CNK2 localization and its function in migration is regulated by AXL. To address these points, we first tested the ability of U2OS cells to migrate randomly and individually. In our experiments, U2OS cells did not migrate extensively either on tissue-culture-treated plates or on a thin Matrigel coating (not shown). We therefore used the Boyden chamber migration and invasion assays, also referred to as transwell assays, to test the role of CNK2 during single cell motility along a chemotactic gradient. Using this assay, we found that the migration and invasion induced by 10% FBS or by 200 ng/mL of recombinant GAS6 was strongly reduced by knockdown of CNK2, indicating that CNK2 is required for single cell chemotaxis toward growth factors, including GAS6 (**Revised Fig. 6h,i; Revised Supplementary Fig. 9d, e**). We report these new results on **lines 374-379**.

The Methods section has also been modified accordingly to account for the new reagents and experimental procedures.

Finally, the experimental metastasis assays are convincing. Yet it remains entirely unclear whether CNK2 is specifically deregulated in solid tumors, or is simply required for basic migratory/invasive activity.

Indeed, we do not currently know whether CNK2 expression is deregulated in solid tumors, which could identify it as a relevant metastasis biomarker and/or drug target. This is an important question that we would like to address in the future, using for instance tumor microarrays probed with our CNK2 antibody. However, such investigations are time consuming and currently beyond the scope of this study.

Reviewer #3 (Remarks to the Author):

The manuscript entitled “The scaffold protein CNK2 promotes cancer cell motility by mediating ARF6 activation downstream of AXL signalling” characterizes the scaffold proteins of the CNK family from human cells, and identifies a role of CNK2 in promoting cancer cell migration through coupling of AXL to the activation of ARF6 GTPase, including a novel adaptor protein called SAMD12. This activation of ARF6-GTP controls motile forces through RAC1 and RHOA GTPases. Mouse xenograft models with CNK2 and SAMD12 KO reduces metastasis, confirming the importance of this pathway in motility of cancer cells and potentially open new avenues for targeting metastasis. The paper thus uncover a previously uncharacterized mechanism of regulation cell motility through a family of scaffold proteins that were poorly characterized up to now.

The authors initially defined the proximal interactome of four human CNKs, in both inducible 293 and U2OS cells using BioID. They identified several interactions including known interactors. Clustering and GO annotations on biological processes revealed enrichments in cell junction organization, cytoskeleton organization, cell polarity and adhesion, although these enrichments are not shown (lanes 88-91), neither is the network of candidates in U2OS cells (lanes 138-139). This should be provided as a table ranked with the enrichment score to have a better view of the functions of the proteins identified. A network with known interactions should also be provided to visualized known interactions in these proteins.

We acknowledge the lack of GO term enrichments for the BioID hits and network representation of the CNK2 proximal interactome. To address this, we have resubmitted our mass spectrometry data for GO term analysis and have updated the dot plots for both BioID experiments conducted in HEK293 and U2OS cells (**Revised Fig. 1m; Revised Supplementary Fig. 1d**). Additionally, we have provided Tables containing the false discovery rate (FDR) for each selected GO term (**Revised Supplementary Fig. 1e; Revised Supplementary Fig. 3d**) that we used to classify the proximal interactors in the dot plot. Finally, we provide a network representation of CNK2 and its proximal interactors from U2OS cells using STRING (version 11.5) (**Revised Supplementary Fig. 3e**). The text was modified to account for these changes (**lines 91-92 & lines 646-647**).

The authors then focus on CNK2 based on the high levels of expression of this gene in a number of cancer cells, including two osteosarcoma cell lines despite lack of expression in normal osteoblast cells. They then characterized how CNK2 regulates RAC1 and RHOA GTPases during migration in a number of assays, and demonstrate that CNK2 operates at the plasma membrane through a phosphoinositide-binding PH domain.

Overall, the authors have demonstrated a CNK2-dependant signalling axis triggered by

AXL that is involved in cancer cell migration, invasion and metastasis. This is a significant discovery and characterization of an important pathway.

We thank Reviewer #3 for appreciating the significance of our work.

One interesting question that remains is whether any of these proteins are direct targets for phosphorylation by the RTK activated (or other kinases), and what other proteins found in proximity are regulated in these conditions. The authors potentially already have this data, since the proteomic analyses included the S/T/Y phosphorylation sites. Since this data has already been extracted, it should at least be mentioned if any relevant modifications were identified. The phosphopeptides information is not provided.

This is an interesting point regarding the phosphopeptides recovered in our BioID experiments. Indeed, it is possible that this data may shed light on potential phospho-dependent regulatory mechanisms. However, we have not thoroughly analyzed this data yet and would therefore respectfully request to leave it for a future study as it is slightly out of topic for this present work.

Other minor suggestions:

Some details are missing from the methods section. In particular, which uniprot database was used? Only the number of entries is provided. The raw files and data should be submitted in a repository for access.

Thank you for raising this concern. We have added the Uniprot release date to the Methods section (**line 626**). Additionally, we have submitted our mass spectrometry datasets to the ProteomeXchange repository and added the identifier (**PXD038687**) (**line 631**).

Some of the figures are very small and difficult to see without extensive zooming of the PDF. For example, the contrast is almost impossible to see in Figure 1d, but much easier to see in Figure 1k.

Indeed, the contrast differs between **Fig. 1d** and **Fig. 1k**. This is due to the use of different microscopes (LSM700 versus Incucyte S3) and imaging software to produce these images. For the initial submission, the file size of the documents was limited, but the resolution would be improved for the final images to be published.

Lane 43: there may be a reference missing with reference 6?

Reference 6 appears on line 43 as (ref. ⁶). This is due to formatting guidelines which don't allow the addition of a superscript at the end of a sentence ending with a number.

Reviewer comments, second round

Reviewer #1 (Remarks to the Author):

The authors have done an excellent job of addressing my earlier concerns. No further changes to the manuscript are necessary.

Reviewer #2 (Remarks to the Author):

The authors performed a set of new experiments to address all my previous comments and suggestions fully.

I have no further issues or clarification.

Reviewer #3 (Remarks to the Author):

The authors have revised Figures 1m, Suppl Fig.1d and have now provided tables containing the requested information in the first round of review, and provided a network representation of CNK2 as suppl figure 3e. I believe this address my main concern regarding this analysis.

Additional information regarding the methods section and references has been corrected as suggested.

The issue on phosphorylation could have been addressed easily from my point of view and would have added interesting data, but the authors have decided to keep this study for the future.

I believe my main concerns have now been addressed, and I would recommend the manuscript for publication.